# Intestinal goblet cells sample and deliver lumenal antigens by regulated endocytic uptake and transcytosis

**Jenny K Gustafsson[1,2]\*, Jazmyne E Davis[2], Tracy Rappai[3], Keely G McDonald[2], Devesha H Kulkarni[2], Kathryn A Knoop[2], Simon P Hogan[4], James AJ Fitzpatrick[3,5,6,7], Wayne I Lencer[8,9,10], Rodney D Newberry[2]\***

[1]Department of Neuroscience and Physiology, University of Gothenburg, Gothenburg, Sweden; [2]Department of Internal Medicine, Washington University School of Medicine, St Louis, United States; [3]Center for Cellular Imaging, Washington University School of Medicine, St Louis, United States; [4]Mary H. Weiser Food Allergy Center, University of Michigan School of Medicine,, Ann Arbor, United States; [5]Department of Cell Biology &Physiology, Washington University School of Medicine, St Louis, United States; [6]Department of Neuroscience, Washington University School of Medicine, St Louis, United States; [7]Department of Biomedical Engineering, Washington University in St. Louis, St. Louis, United States; [8]Department of Pediatrics, Harvard Medical School, Boston, United States; [9]Division of Gastroenterology, Nutrition and Hepatology, Boston Children's Hospital, Boston, United States; [10]Harvard Digestive Disease Center, Harvard Medical School, Boston, United States

**\*For correspondence:**
jenny.gustafsson@gu.se (JKG);
rnewberry@wustl.edu (RDN)

**Competing interest:** The authors declare that no competing interests exist.

**Abstract** Intestinal goblet cells maintain the protective epithelial barrier through mucus secretion and yet sample lumenal substances for immune processing through formation of goblet cell associated antigen passages (GAPs). The cellular biology of GAPs and how these divergent processes are balanced and regulated by goblet cells remains unknown. Using high-resolution light and electron microscopy, we found that in mice, GAPs were formed by an acetylcholine (ACh)-dependent endocytic event remarkable for delivery of fluid-phase cargo retrograde into the trans-golgi network and across the cell by transcytosis – in addition to the expected transport of fluid-phase cargo by endosomes to multi-vesicular bodies and lysosomes. While ACh also induced goblet cells to secrete mucins, ACh-induced GAP formation and mucin secretion were functionally independent and mediated by different receptors and signaling pathways, enabling goblet cells to differentially regulate these processes to accommodate the dynamically changing demands of the mucosal environment for barrier maintenance and sampling of lumenal substances.

## Editor's evaluation

In this study, the authors present a compelling data regarding how barrier function and immunity are coordinated within the mammalian intestinal system. This paper demonstrates that cells responsible for the secretion of the protective mucous lining of the intestine also sample substances within the intestinal lumen to present to the immune system, and how these two processes are differentially regulated.

## Introduction

The simple columnar epithelium lining the gastrointestinal tract is an expansive surface exposed to lumenal contents containing innocuous substances from the diet and potentially harmful microbes and their products. Underneath and within this epithelium lies the largest collection of immune cells in the body. It has long been appreciated that the gastrointestinal immune system is not ignorant of the lumenal contents, and in the healthy state gut lumenal contents are sampled to induce adaptive immune responses, characterized by T cell-mediated antigen-specific tolerance. How substances from the lumen are encountered by, or delivered to, the immune compartment is a central concept in mucosal immunity that is incompletely understood, but fundamentally underlies how the immune system can mount tolerogenic responses to substances encountered in this potentially hostile environment.

Goblet cells (GCs) are specialized intestinal epithelial cells that have a well-established role in innate immunity through secretion of mucins and maintenance of the mucus layer. The mucus layer provides a first line of defense against physical and chemical injury and protects against pathogen invasion (*Kim and Ho, 2010*). GCs also produce and secrete biologically active products that contribute to innate immunity by promoting epithelial restitution, inhibiting intestinal nematode chemotaxis, and stabilizing the mucus layer (*Herbert et al., 2009*; *Johansson et al., 2009*; *Taupin and Podolsky, 2003*). Recently, a new role for GCs in intestinal immunity was identified; the ability of GCs to form GC-associated antigen passages (GAPs), which deliver lumenal substances to lamina propria antigen presenting cells to generate antigen-specific T cell responses (*Knoop et al., 2017a*; *Knoop et al., 2017b*; *Knoop et al., 2015*; *McDole et al., 2012*; *Knoop et al., 2017b*). The potent GC secretagogue acetylcholine (ACh) induces GAP formation in the homeostatic state (*Knoop et al., 2015*; *Miller et al., 2014*), indicating that GAP formation and mucus secretion can be induced by the same stimulus. However, GAP formation and the delivery of lumenal substances to the immune compartment are closely regulated to prevent inflammatory responses to gut bacteria and dietary antigens in settings where the secretion of mucus to maintain the barrier is critical (*Knoop et al., 2017a*; *Knoop et al., 2016*; *Kulkarni et al., 2018*). Thus, in response to the same stimulus, GCs perform two, but apparently opposing, central roles in intestinal immunity.

Here, we demonstrate that GAPs form by an endocytic event capturing fluid-phase cargo and often follow secretory granule exocytosis. But in contrast to the fate of fluid-phase endocytosis seen in adjacent enterocytes, GAPs traffic a substantial portion of fluid-phase cargo across the epithelial barrier by transcytosis for capture by underlying phagocytic cells. Further, we find that while ACh induces both mucus secretion and GAP formation in GCs, these processes are not functionally linked and can be performed independently or in parallel to meet the needs of the changing lumenal environment in the small intestine and colon. Together these observations suggest that GAP formation may have evolved as a cell-type specific and specialized endocytic pathway emerging from the basic exocytic machinery of GCs to sample and deliver luminal substances to the mucosal immune system.

## Results

### GAP formation is a fluid-phase endocytic process dependent on PI3K, actin polymerization, and microtubule transport

To elucidate the cellular mechanism underlying GAP formation and function, we studied the intestines of mice 1 hr after administration of fluorescent dextran to the gut lumen. The appearance of GAPs on two-photon in vivo imaging (*Figure 1A*) and on wide-field fluorescence microscopy (*Figure 1B*) of the small intestine (SI) suggested that the GC cytoplasm was filled with lumenal substances (marked by fluorescently labeled dextran), consistent with passive diffusion of dextran into the cytosol through apical membrane tears following ACh-induced mucus granule secretion. To better understand this process, we examined GAPs using super-resolution microscopy approaches. The appearance of SI GAPs by structured illumination microscopy (SIM) revealed that the cytoplasm of GCs forming GAPs was not completely filled with the fluorescent luminal cargo, rather the luminal solute tracer was contained within a network of vesicular appearing structures located predominantly at the periphery of the cell and extending from apical to basolateral cell surfaces, but largely excluded from the region where mucin granules are contained within the theca. Notably, in places, the luminally administered ovalbumin (OVA) was also observed in vesicular appearing structures contained within Itgax[YFP+]

**eLife digest** Cells in the gut need to be protected against the many harmful microbes which inhabit this environment. Yet the immune system also needs to 'keep an eye' on intestinal contents to maintain tolerance to innocuous substances, such as those from the diet. The 'goblet cells' that are part of the gut lining do both: they create a mucus barrier that stops germs from invading the body, but they also can pass on molecules from the intestine to immune cells deep in the tissue to promote tolerance. This is achieved through a 'GAP' mechanism. A chemical messenger called acetylcholine can trigger both mucus release and the GAP process in goblet cells.

Gustafsson et al. investigated how the cells could take on these two seemingly opposing roles in response to the same signal. A fluorescent molecule was introduced into the intestines of mice, and monitored as it pass through the goblet cells. This revealed how the GAP process took place: the cells were able to capture molecules from the intestines, wrap them in internal sack-like vesicles and then transport them across the entire cell.

To explore the role of acetylcholine, Gustafsson et al. blocked the receptors that detect the messenger at the surface of goblet cells. Different receptors and therefore different cascades of molecular events were found to control mucus secretion and GAP formation; this explains how the two processes can be performed in parallel and independently from each other.

Understanding how cells relay molecules to the immune system is relevant to other tissues in contact with the environment, such as the eyes, the airways, or the inside of the genital and urinary tracts. Understanding, and then ultimately harnessing this mechanism could help design of new ways to deliver drugs to the immune system and alter immune outcomes.

mononuclear phagocytes (MNPs) located beneath the epithelial barrier (*Figure 1C*). The vesicular patterns observed suggested that GAPs were formed by an active endocytic process that delivered fluid-phase cargo across the epithelial barrier to the lamina propria. Indeed, blockade of endocytosis by the dynamin inhibitors Dyngo 4 a and dynasore strongly attenuated GAP formation (*Figure 1D* and *Figure 1—figure supplement 1A–C*). Inhibition of actin and microtubule polymerization using cytochalasin D and colchicine, respectively, and inhibition of phosphoinositide three kinase (PI3K) using LY294002 also resulted in a significant reduction in GAP formation (*Figure 1D* and *Figure 1—figure supplement 1D–F*), as did inhibition of the microtubule motor proteins dynein and kinesin using ciliobrevin D and dimethylenastron (DMEA) respectively (*Figure 1D* and *Figure 1—figure supplement 1G,H*). These treatments did not adversely affect epithelial integrity (*Figure 1—figure supplement 1A–H*). Thus, GAPs appear to represent a fluid-phase endocytic pathway capable of delivering lumenal solutes across the cell by transcytosis (*Clague et al., 1995*; *Gottlieb et al., 1993*; *Thyberg and Stenseth, 1981*).

The endocytic capacity of secretory cells such as neurons and enterocromaffin cells has been extensively studied using lipophilic styryl dyes such as FM 1–43 FX (*Pyle et al., 1999*; *Raghupathi et al., 2016*). These dyes become fluorescent when incorporated in membranes, and the endocytic capacity of a cell and the intracellular fate of the endocytic vesicles can be studied by evaluating the subcellular localization of FM 1–43 FX stained membranes. Lumenal administration of FM 1–43 FX resulted in intracellular staining of a subset of GCs, with a staining pattern similar to that of endocytosed dextran (*Figure 1E* compare with *Figure 1B*). To evaluate whether endocytosed dextran localized to newly endocytosed membrane structures within the same cell, FM 1–43 FX and 10 kDa dextran-Alexa647 were co-administered into the SI lumen and the intracellular staining pattern was evaluated by confocal microscopy. We observed a similar staining pattern of dextran-Alexa647 and FM 1–43 FX within the same cell having GC morphology (*Figure 1F*), further supporting that GAP formation represents an endocytic process. In addition to observing intracellular FM 1–43 FX staining and dextran uptake by GCs, we also observed uptake of lumenal dextran by surrounding intestinal epithelial cells. However, the pattern and morphology of dextran-containing endosomes in absorptive cells differed from those in GCs. In GCs (asterisk; *Figure 1G* and upper panel insets), dextran-containing structures were observed throughout the cell, whereas in adjacent epithelial cells, likely enterocytes (arrows; *Figure 1G* and lower panel insets), a punctuated pattern of FM 1–43 FX staining and dextran uptake was observed localized to the apical cytoplasm. In adjacent epithelial cells, we

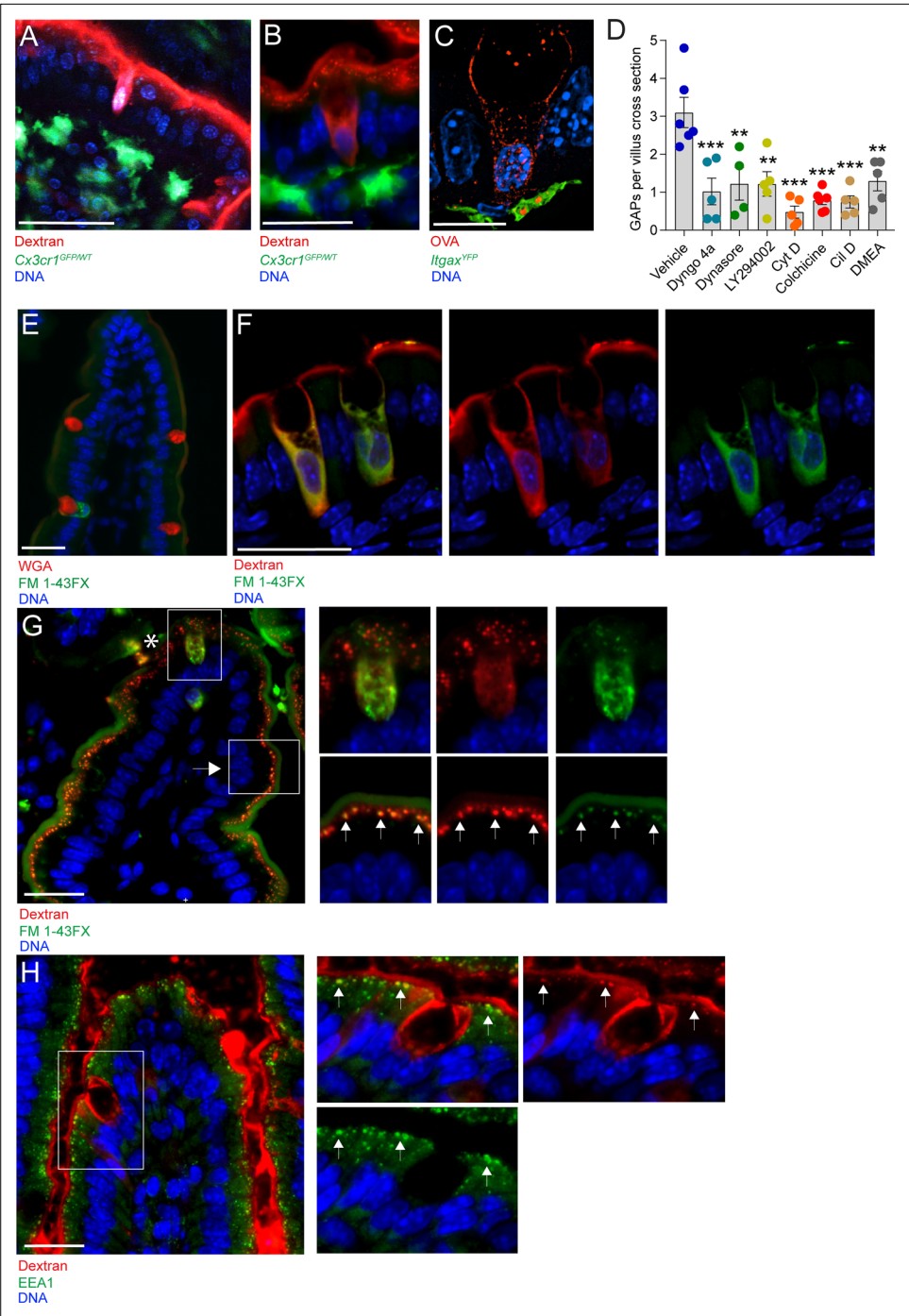

**Figure 1.** Goblet cell associated antigen passage (GAP) formation is an active endocytic process dependent on actin polymerization, microtubule transport, and phosphoinositide three kinase (PI3K). (**A**) Intravital two-photon imaging, (**B**) wide-field fluorescent imaging, and (**C**) super-resolution structured illumination microscopy (SIM) imaging of the small intestine (SI) of CX3CR1[GFP] (green) or Itgax[YFP] (green) reporter mice following luminal administration of 10 kDa tetramethylrodamine (TRITC)-dextran (red) or Texas Red-Ovalbumin (OVA; red). (**D**) Quantification of GAPs per SI villus cross section in mice treated with vehicle (n = 6), Dyngo 4a (n = 5), dynasore (n = 4), LY294002 (n = 5), cytochalasin D (Cyt D, n = 5), colchicine (n = 6), ciliobrevin D (Cil D, n = 5), or dimethylenastron (DMEA, n = 5) followed by intraluminal administration of 10 kDa TRITC-dextran. (**E**) Wide-field fluorescent imaging of the SI following intraluminal administration of FM 1–43 FX (green) for 1 hr. Goblet cells are visualized by wheat germ agglutinin (WGA-Texas Red) (red). (**F**) Confocal fluorescent imaging of the SI following intraluminal administration of FM 1–43 FX (green) and dextran-Alexa647 (red). (**G**) Wide-field fluorescent imaging

*Figure 1 continued on next page*

*Figure 1 continued*

of the SI following intraluminal administration of FM 1–43 FX (green) and dextran-Alexa647 (red). Asterisk denotes goblet cell, arrow denotes enterocyte, arrows in zoomed in pictures denotes endosomal structures double positive for FM 1–43 FX and dextran-Alexa647 in enterocytes. (**H**) Wide-field fluorescent imaging of the SI following intraluminal administration of TRITC-dextran (red) stained for early endosome protein 1 (EEA1) (green). Arrows denote endosomal structures double positive for TRITC-dextran and EEA1 in enterocytes. Data are presented as mean ± SEM. **p < 0.01, ***p < 0.001 as compared to vehicle treated mice. Scale bar: (A) 50 µm, (**B**) 25 µm, (**C**) 10 µm, (**E–H**) 25 µm. (**C** and F) represent 3D projections of obtained z-stacks. Statistical analysis was performed using a one-way ANOVA followed by Dunnet's post-hoc test. A–B, E, G–H n = 6, C, F n = 4. Each point in panel D represents the average number of GAPs from 25 villi in one mouse.

The online version of this article includes the following figure supplement(s) for figure 1:

**Figure supplement 1.** Effect of goblet cell-associated antigen passage (GAP) inhibition on tissue integrity.

also observed co-localization of dextran-containing vesicles and the early endosome marker EEA1 (arrows; *Figure 1H*), but EEA1 staining was not apparent in the endocytic vesicles of adjacent GCs. Thus, the vesicles and trafficking pathways of fluid-phase endocytosis differ between GCs forming GAPs and adjacent enterocytes.

## An ultrastructural model suggests GAPs form by recovery of secretory granule membranes which traffic fluid-phase cargo to the trans-Golgi network (TGN) and across the cell by transcytosis in addition to the lysosome

To understand the trafficking and fates of the vesicles-containing lumenal cargo in GAPs, we performed focused ion beam scanning electron microscopy (FIB-SEM) using intestinal tissue obtained from mice 1 hr after administration of 10 kDa lysine fixable dextran biotin into the gut lumen. Full thickness tissues were incubated with heavy metal labeled streptavidin and processed for FIB-SEM and imaged at a voxel resolution of 10 nm per slice with >1000 slices encompassing the thickness of a GC. Datasets of four GCs endocytosing lumenal dextran were imaged with this technique. A dataset containing 1199 images encompassing the volume of a GC that had taken up dextran was used to create the 3D model of a GC forming a GAP illustrated in *Figure 2A* and *Video 1*. The remaining datasets as well as transmission electron microscopy (TEM) of dextran incubated tissues were used as to confirm the model. The endocytosed dextran was seen in apically located endosomes, multi-vesicular bodies (MVBs), lysosomes, the TGN, and in vesicles close to basolateral membrane consistent with delivery of dextran into the transcytotic pathway (*Figure 2A and B* and *Video 2*). Shuttling of endocytosed cargo to endo-lysosomal structures as well as the TGN is consistent with models of fluid-phase endocytosis and their trafficking in most cell types (*Maxfield and McGraw, 2004*), and the specialized need for compensatory endocytosis required by secretory cell types like GCs to recycle large amounts of membrane back into the secretory pathway (i.e., TGN) after exocytosis (*Engisch and Nowycky, 1998*). Notably the endocytic vesicles in GCs forming GAPs did not stain positive for EEA1 (*Figure 1H*), a marker of early endosomes, or the late endosome marker Rab7 (*Figure 2—figure supplement 1A*), but did in some cells stain positive for the lysosome marker LAMP-1 and the TGN marker TGN46 (*Figure 2—figure supplement 1B, C*). To explore if the endosomes of GAPs could be related to membrane retrieval following secretory granule exocytosis, we examined the location of the secretory granule protein Rab3D that has been shown to be retrieved and redistributed to the TGN following regulated secretion in GCs and pancreatic acinar cells (*Jena et al., 1994*; *Valentijn et al., 2007*). We found that in GCs forming GAPs, Rab3D co-localized with areas of dextran uptake, while in GCs not forming GAPs, Rab3D was localized to the secretory granules within the GC theca (*Figure 2C* and compare with *Figure 2D*). A similar staining pattern was observed when evaluating the localization of the secretory granule protein VAMP8 in GCs forming GAPs (*Figure 2—figure supplement 1D*). These results could be consistent with GAP endosomes forming as a result of membrane recycling after secretory granule exocytosis, which was supported by findings of dextran localizing to wheat germ agglutinin (WGA)-positive vesicular structures in GCs forming GAPs (*Figure 2E*). Furthermore, unlike absorptive intestinal epithelial cells where transcytosis is largely restricted to receptor-mediated endocytosis (*Fung et al., 2018*), the GCs forming GAPs appeared to deliver fluid-phase cargo into the transcytotic pathway, as evidenced by dextran-containing vesicles closely adjacent to the basolateral

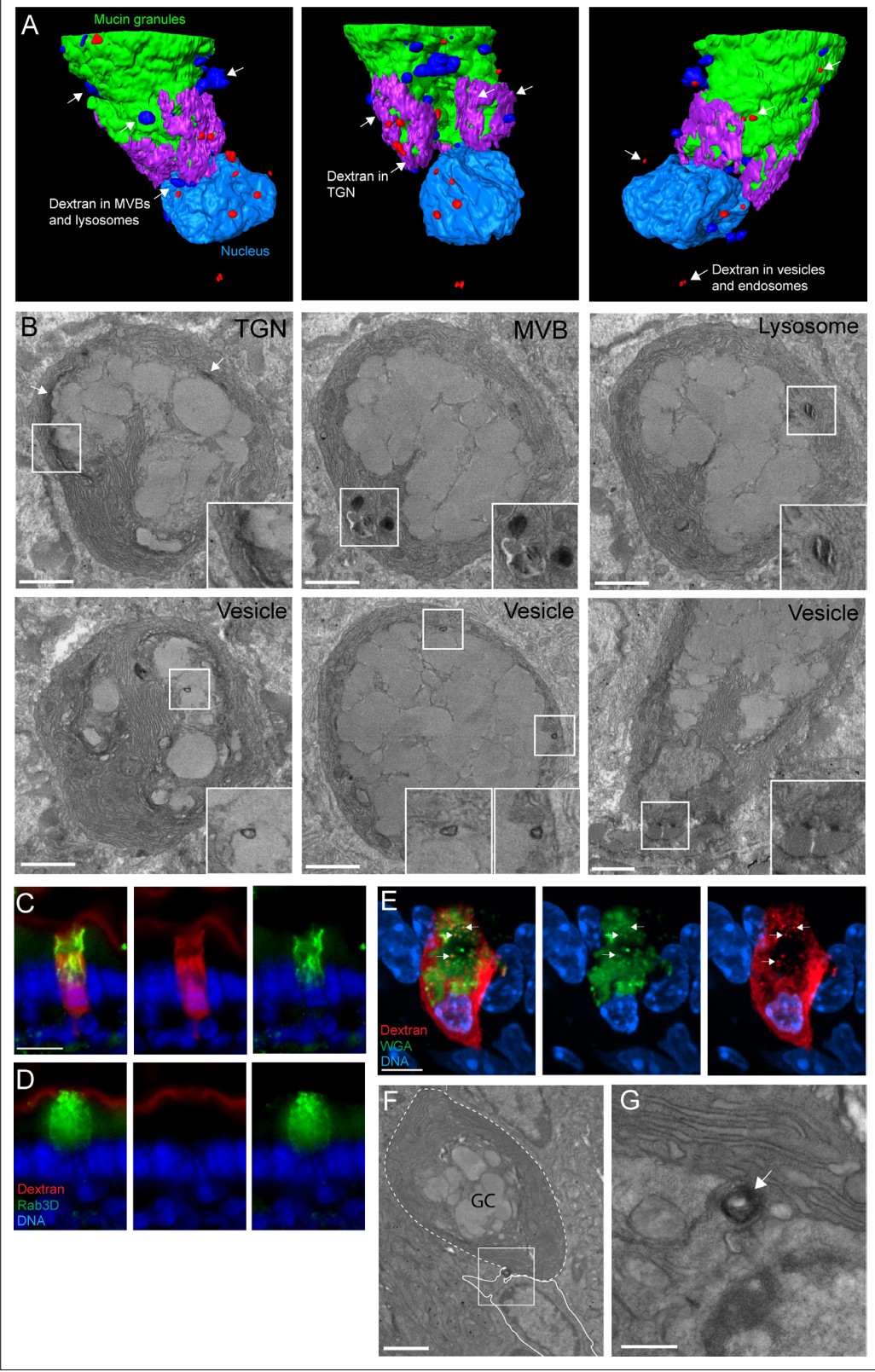

**Figure 2.** An ultrastructural model of goblet cell (GC)-associated antigen passage (GAP) formation. (**A**) 3D model of the compiled focused ion beam scanning electron microscopy (FIB-SEM) data demonstrating the fates of luminal cargo in a GAP. Green: mucin granules, purple: dextran in the trans-Golgi network (TGN), dark blue: dextran in multi-vesicular bodies (MVBs) and lysosomes, red: dextran in vesicles/endosomes, light blue: nucleus.

*Figure 2 continued on next page*

*Figure 2 continued*

(**B**) Representative FIB-SEM images showing dextran localizing to TGN, MVBs, lysosomes, and endosomes/vesicles. (**C and D**) Wide-field fluorescent imaging of the small intestine (SI) following intraluminal administration of tetramethylrodamine (TRITC)-dextran (red) and stained for Rab3D (green), and 4',6-diamidino-2-phenylindole (DAPI) blue (DNA). (**C**) Goblet cells (GCs) forming GAPs show redistribution of Rab3D (green) with uptake of lumenal dextran. (**D**) GCs not forming GAPs show Rab3D localized to secretory granules in the theca. (**E**) Confocal fluorescent imaging of SI following intraluminal injection of TRITC-dextran (red) stained with wheat germ agglutinin (WGA) (green) and DAPI (blue). Arrows denote vesicular structures double positive for dextran and WGA. (**F** and **G**) Transmission electron micrographs of the basolateral area of a GC following administration of intraluminal dextran. The dashed line outlines a GC and the solid outlines an adjacent underlying cell. Arrow denote a dextran-containing structure located at the fusion point of the GC and the adjacent cell. Scale bar: (B and F) 2 µm, (**C–D**) 10 µm, (**E**) 5 µm, (**G**) 500 nm. n = 4 in panel B, n = 5 in panel C, D, E, n = 3 in panel E, F.

The online version of this article includes the following figure supplement(s) for figure 2:

**Figure supplement 1.** Goblet cell-associated antigen passage (GAP) formation is associated with transport of cargo to the lysosome and the trans-Golgi network (TGN).

membranes (*Figure 2B*). In some cases, dextran-containing cargo was observed at a fusion point of GCs and adjacent cells in the lamina propria (*Figure 2F and G*), which is consistent with the transfer of luminal cargo taken up by GAPs to lamina propria MNPs we observed by SIM (*Figure 1C*). These results suggest that GAPs represent a fluid-phase transcellular endocytic process capable of efficiently delivering lumenal substances across the epithelial barrier to be captured by phagocytic cells of the innate immune system.

## Steady-state GAP formation is regulated by ACh acting on mAChR4 in the SI and mAChR3 in the distal colon

We have previously shown that in the homeostatic state GAPs are present in the distal colon and SI, preferentially occurring in villus GCs, but largely absent in the proximal colon due to GC intrinsic sensing of the microbiota inhibiting muscarinic ACh receptor 4 (mAChR4)-driven GAP formation in adult mice (*Knoop et al., 2015*; *Kulkarni et al., 2020*). In the SI, steady-state GAP formation is mediated by ACh acting on mAChR4 expressed by GCs (*Knoop et al., 2015*). ACh is a potent GC secretagogue known to induce secretion of large quantities of mucus by activation of mAChRs when added to intestinal explants. However, baseline mucus secretion in the intestine has not been linked to muscarinic receptor activation (*Specian and Neutra, 1980*), suggesting that during steady state, activation of mAChR4 triggers endocytic retrieval of secretory granule membrane without being the main driver of secretory granule exocytosis. To explore this further we treated mice with the mAChR4 antagonist tropicamide and evaluated the effect on GAP formation and the mucus remaining within the GCs as an indication of inhibition of

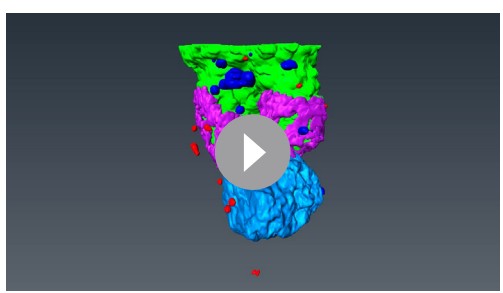

**Video 1.** Animation of the 3D model of the focused ion beam scanning electron microscopy (FIB-SEM) data. *Green: Mucin granules, light blue: nucleus, red: dextran in vesicular and endosomal structures, dark blue: dextran in lysosomes and multi-vesicular bodies (MVBs) and purple: dextran in the trans-Golgi network (TGN).*
https://elifesciences.org/articles/67292/figures#video1

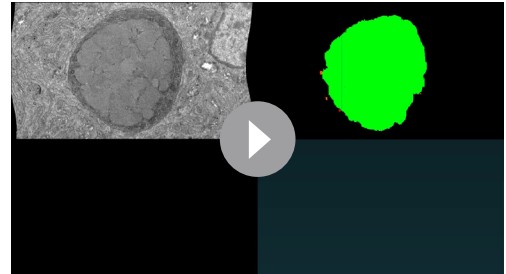

**Video 2.** Animation of the focused ion beam scanning electron microscopy (FIB-SEM) pictures and the corresponding segmentation that the model is based on. *Upper left panel: FIB-SEM pictures, upper right panel: segmentation of the nucleus (blue), mucin granuels (green), dextran in endosomes/vesicles (red), dextran in MVBs and lysosomes (orange). Lower left panel: segmentation of dextran in the trans-Golgi network (TGN) (blue).*
https://elifesciences.org/articles/67292/figures#video2

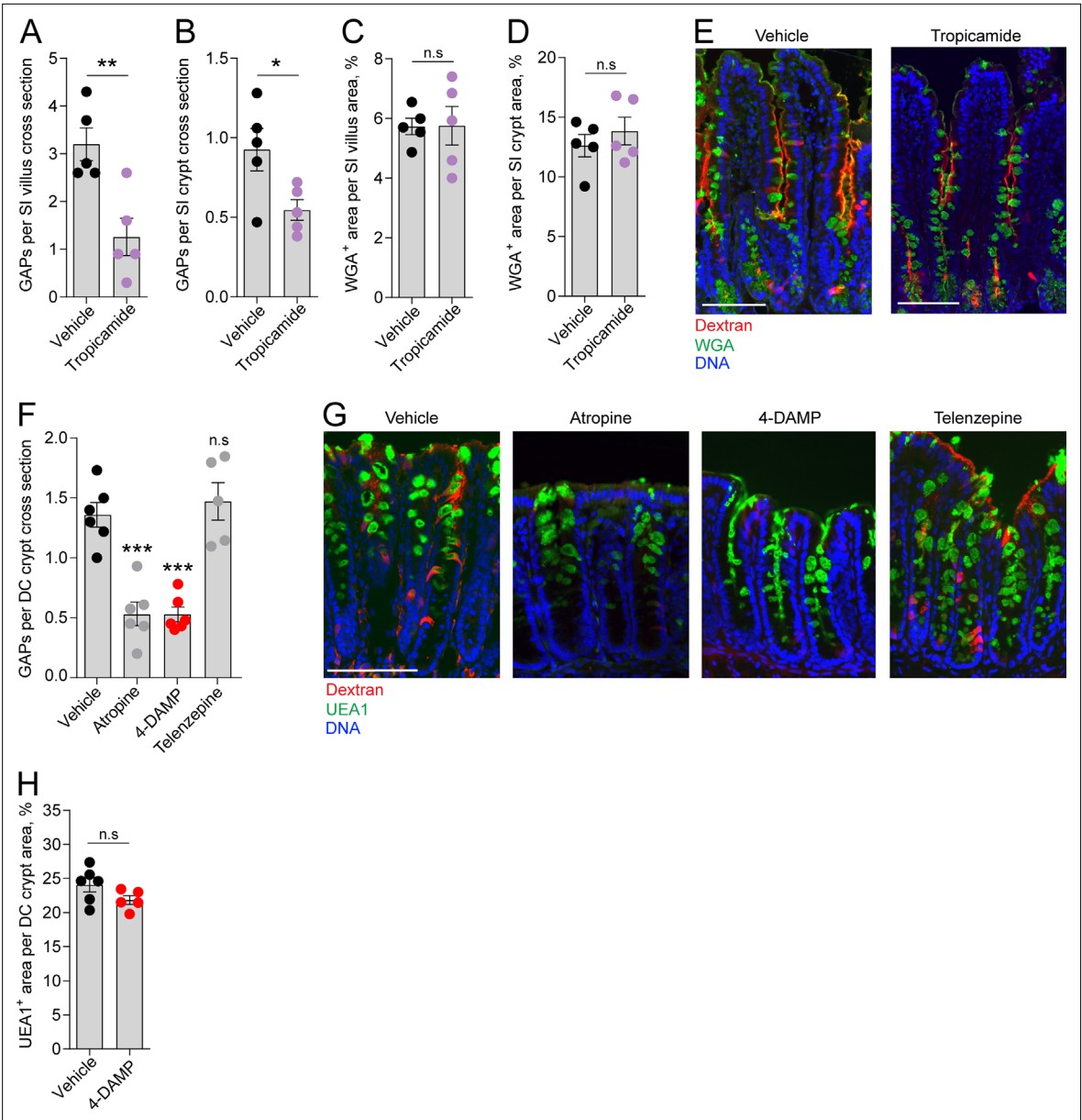

**Figure 3.** Role of muscarinic acetylcholine receptors (mAChRs) in steady-state goblet cell-associated antigen passage (GAP) formation and mucus secretion in the small intestine (SI) and distal colon. GAP numbers per (**A**) SI villus and (**B**) SI crypt in mice treated with vehicle or the mAChR4 antagonist tropicamide. Mucus content of the (**C**) SI villus and (**D**) SI crypt quantified as percentage of wheat germ agglutinin-positive (WGA⁺) area per villus/crypt area in mice treated with vehicle or tropicamide. (**E**) Representative wide-field fluorescent image of the SI of mice treated with vehicle or tropicamide following intraluminal administration of tetramethylrodamine (TRITC)-dextran (red) and WGA staining of mucus (green). (**F**) GAP numbers per distal colon (DC) crypt in mice treated with vehicle, pan muscarinic receptor antagonist atropine, mAChR3 antagonist 1,1-dimethyl-4-diphenylacetoxypiperidinium iodide (4-DAMP), or mAChR1 antagonist telenzepine. (**G**) Representative wide-field fluorescent imaging of distal colon crypts of mice treated with vehicle, atropine, 4-DAMP, or telenzepine following intraluminal administration of TRITC-dextran (red) and *Ulex europaeus* agglutinin 1 (UEA1) staining of mucus. (**H**) Mucus content of the distal colon crypt of mice treated with vehicle or 4-DAMP, quantified as percentage of UEA1⁺ area per distal colon crypt area. Data are presented as mean ± SEM. *p<0.05, **p<0.01, ***p<0.001 as compared to vehicle. n.s. = non-significant as compared to vehicle. n = 5 in panels **A–D**, n = 6 in panel **F** (telenzepine n = 5). Each data point in **A–D, F, H** represents the average of 25 villi or 40 crypts from one mouse. Statistical analysis was performed using an unpaired two-sided Student's t-test in panel **A–D, H**. A one-way ANOVA followed by Dunnet's post hoc test was performed in panel **F**.

baseline mucus secretion. Our results showed that inhibition of mAChR4 signaling by tropicamide reduced GAP formation in the SI villus and crypt as expected (*Figure 3A and B*, image E), but had no measurable effect on mucus secretion, quantified as WGA$^+$ mucus area per villus or crypt cross section (*Figure 3C and D*, image E).

We have previously shown that GAP formation in the distal colon is driven by ACh acting on muscarinic receptors, but independent of mAChR4 signaling (*Kulkarni et al., 2020*). However, colonic GCs have been shown to express mAChR3 and mAChR1 (*Tabula Muris Consortium et al., 2018*). 1,1-Dimethyl-4-diphenylacetoxypiperidinium iodide (4-DAMP) is an mAChR3 preferring antagonist with some capacity to bind mAChR1. Treatment of mice with 4-DAMP significantly decreased distal colon GAP formation similar to that observed with the pan-muscarinic receptor antagonist atropine, while the mAChR1 antagonist telenzepine had no measurable effect on GAP formation (*Figure 3F*, image G), indicating that distal colonic GAPs are driven by mAChR3 signaling. Similar to the SI, inhibition of mAChR3 signaling using 4-DAMP had no measurable effect on mucus secretion in the distal colon as evaluated by *Ulex europaeus* agglutinin 1 (UEA1$^+$) mucus area per crypt cross section (*Figure 3H*). Thus, mAChR4 and mAChR3 drive GAP formation in the SI and distal colon, respectively, but do not play a major role in baseline mucus secretion.

## Exposure to high concentrations of ACh induces a response favoring GAP formation in the SI villus and mucus secretion in the crypts

Our results indicate that in the steady state, ACh acting on mAChR4 in the SI and mAChR3 in the distal colon triggers a response resulting in GAP formation with negligible effects on the degree of mucus secretion (*Figure 3A–H*). In contrast, exposing intestinal explants to micro-molar concentrations of ACh induces rapid expulsion of large quantities of mucus, primarily from the intestinal crypts (*Ermund et al., 2013*; *Gustafsson et al., 2012*; *Phillips, 1992*; *Specian and Neutra, 1980*). To explore how GAP formation correlates with mucus secretion induced by the same stimulus, ACh, we treated mice with the stable ACh analogue carbamylcholine (CCh) and measured GAP formation and mucus secretion. Mucus secretion was assayed as a decrease in mucus content within the tissue. Exposure to CCh resulted in a significant increase in GAP formation in the SI villus and crypts and the distal colon crypts (*Figure 4A*, images C, F, and G) and was paralleled by an accelerated mucus secretory response defined as loss of WGA$^+$ (SI) or UEA1$^+$ (distal colon) mucus area, at all three locations (*Figure 4B*, images C, F, G). Despite all three locations responding with a mucus secretory response, the degree and type of response differed between compartments. In the crypts of the SI and distal colon, but not the SI villus, exposure to CCh resulted in a significant reduction in the number of WGA$^+$ (SI) and UEA1$^+$ (distal colon) GCs (*Figure 4D*), indicative of complete emptying of a subset of GCs in these regions of the alimentary tract – the crypts of the SI and colon. However, the remaining SI crypt GCs that did not undergo complete emptying and SI villus GCs demonstrated a reduction in the GC theca area (*Figure 4E*) consistent with partial emptying of the mucin granules in response to CCh in this compartment. The remaining GCs in the distal colon crypts staining positive for mucins demonstrated no change in the size of the GC theca (*Figure 4E*), suggesting that GCs in the distal colon predominantly responded to CCh by complete emptying of the theca.

Further correlation of the CCh-induced mucus secretory response with the GAP response between compartments revealed that the SI crypts had the largest reduction of total mucus content per cross section (crypt SI: –47.2% ± 2.2%, villus SI: –34.1% ± 3.1%, crypt DC: –32.8% ± 4.9%, crypt SI vs. villus SI: $p < 0.05$, crypt SI vs. crypt DC: $p < 0.05$), whereas the SI villus had the largest GAP response to CCh quantified as the percentage of GCs forming GAPs in response to CCh (villus SI: 58.8% ± 2.5%, crypt SI: 21.38% ± 1.2%, crypt DC: 17.4% ± 1.2%, villus SI vs. crypt SI, $p < 0.001$, villus SI vs. crypt DC: $p < 0.001$, crypt SI vs. crypt DC, $p > 0.05$). The SI and distal colon crypts responded to CCh with complete emptying of an average of 3 and 4 GCs per crypt cross section, respectively (*Figure 4D*), and formed ~1 additional GAP per cross section (*Figure 4A*), demonstrating that the degree of CCh-induced mucus secretion does not directly correlate with the extent of GAP formation. We have previously observed Paneth cells acquiring lumenal administered dextran and ovalbumin (*Kulkarni et al., 2020*; *Noah et al., 2019*) and observed Paneth cells acquiring dextran in the present study (*Figure 4H*), therefore the low frequency of GAP formation in the intestinal crypts cannot be due to restricted access of dextran to the SI crypt lumen. Thus, in response to CCh, GCs in all the examined locations of the intestine produced a mucus secretory response and formed GAPs, however SI and

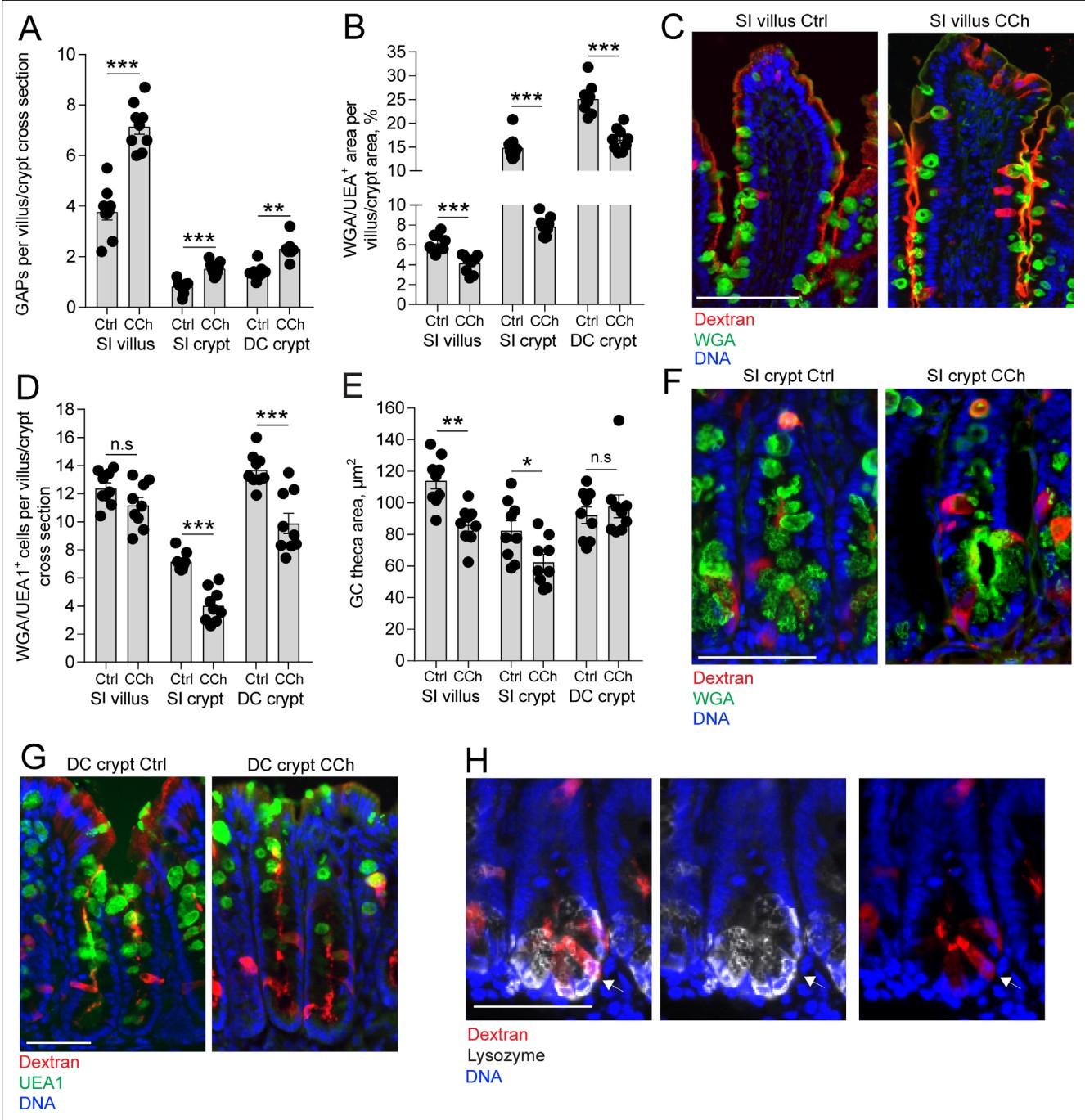

**Figure 4.** Carbamylcholine (CCh)-induced goblet cell (GC)-associated antigen passage (GAP) formation and mucus secretion in the small intestine (SI) and distal colon. (**A**) GAP numbers, (**B**) total wheat germ agglutinin-positive (WGA+) (SI) or *Ulex europaeus* agglutinin 1 (UEA1+) (distal colon) mucus area per villus/crypt area, (**C**) representative wide-field fluorescent image of SI villus, (**D**) quantification of the number of WGA+ (SI) or UEA1+ (distal colon) GCs per villus or crypt cross section, (**E**) quantification of GC theca area, (**F**) representative wide-field fluorescent image of SI crypts in vehicle and CCh treated tissue. (**G**) Representative wide-field fluorescent image of distal colon crypts and (H) representative wide-field fluorescent image of an SI crypt following intraluminal tetramethylrodamine (TRITC)-dextran (red) administration stained for Lysozyme (white). Arrows denote a Paneth cell containing TRITC-dextran. Scale bar: (C) 100 μm, (**F, J, K**) 50 μm. Data are presented as mean ± SEM, *p < 0.05, **p < 0.01, ***p < 0.001, n.s = non-significant, n = 9 in all groups. Statistical analysis was performed using an unpaired two-sided Student's t-test. Each data point in A–B, D–E represents the average of 25 villi or 40 crypts from one mouse.

distal colon crypt GCs were more likely to completely empty their mucus content and villus GCs were more likely to form GAPs, further suggesting that GAP formation and mucus secretion may occur independently of each other, even in response to the same stimulus.

## GCs use separate muscarinic receptors and different Ca²⁺ pathways to perform ACh-induced mucus secretion and ACh-induced GAP formation

To evaluate if the observed regional and spatial differences in ACh-induced mucus secretion and GAP formation were due to the two processes being mediated by different mAChRs, we treated mice with mAChR antagonists and measured the effect on CCh-induced GAP formation and mucus secretion. Our results showed that the mAChR4 antagonist tropicamide had no effect on the CCh-induced mucus secretory response in either the SI villus or crypt compartment (*Figure 5A and C*), but it blocked the CCh-induced GAP response at both locations (*Figure 5B and D*), similar to the effects of mAChR4 blockade in the basal state (*Figure 3A and B*). In the distal colon, the mAChR3 inhibitor 4-DAMP inhibited the CCh-induced GAP response similar to the effects of mAChR3 inhibition in the basal state (*Figure 5F*), but it did not reverse the CCh-induced mucus secretory response (*Figure 5E*) – indicating that the CCh-induced mucus secretory response was mediated by other mAChRs. As previously noted, intestinal GCs also express mAChR1 (*Haber et al., 2017*; *Tabula Muris Consortium et al., 2018*). Evaluation of the role of mAChR1 in regulating the CCh response showed that pretreatment of mice with the mAChR1 antagonist telenzepine reversed both the CCh-induced mucus secretory response (*Figure 5A, C and E*) and the CCh-induced GAP response (*Figure 5B, D and F*) at all three locations. Thus, the CCh-induced mucus secretory response is mediated by activation of mAChR1 while the CCh-induced increase in GAPs involves activation of both mAChR1 and mAChR4 in the SI, and mAChR1 and mAChR3 in the distal colon.

We evaluated the expression of mAChRs in the SI and distal colon to correlate the CCh-induced mucus secretory response and the GAP response with expression of the respective receptors. In the SI, the mAChR4 expression increased along the crypt – villus axis, correlating with the higher prevalence of mAChR4-dependent GAP formation in the villi as compared to the crypts (*Figure 5—figure supplement 1A*). In contrast, the mAChR1 expression was more evenly distributed along the crypt – villus axis, correlating with mAChR1's involvement in both the mucus secretory response and the GAP response in the SI villi and crypts (*Figure 5—figure supplement 1B*). In the distal colon, epithelial expression of mAChR3 was primarily located to the the lower part of the crypts, correlating with the location of distal colon GAPs (*Figure 5—figure supplement 2A*). In addition, mAChR3 expression was also observed in the muscle layers (*Figure 5—figure supplement 2A*). Similar to mAChR3, mAChR1 expression was primarily observed in the lower part of the crypt correlating with the location of the CCh-induced mucus secretory response (*Figure 5—figure supplement 2B*).

Based on previous studies demonstrating that ACh-induced mucus secretion is mediated by elevated levels of intracellular Ca²⁺ and endocytosis being a Ca²⁺-dependent process (*Barbieri et al., 1984*; *Seidler and Sewing, 1989*), we explored the role of Ca²⁺ in regulating ACh-induced mucus secretion and GAP formation. Elevated levels of intracellular Ca²⁺ can either be obtained by Ca²⁺ release from intracellular stores or activation of Ca²⁺ channels in the plasma membrane triggering influx of extracellular Ca²⁺. Therefore, we explored the source of Ca²⁺ needed to drive CCh-induced increase in GAP formation and mucus secretion in the SI and distal colon. Our results showed that the CCh-induced mucus secretory response was reversed by chelation of extracellular Ca²⁺ (EGTA) as well as by chelation of intracellular Ca²⁺ (BAPTA-AM) in the SI (*Figure 6A, B and E–H*) and in the distal colon (*Figure 7A and C–F*), indicating that the CCh-induced mucus secretory response is dependent upon influx of extracellular Ca²⁺ and possibly also dependent on Ca²⁺ release from intracellular stores. In contrast, the CCh-induced GAP response in the SI (*Figure 6C–H*) and distal colon (*Figure 7B–F*) was independent of extracellular Ca²⁺, but dependent on intracellular Ca²⁺, and by extension dependent upon the release of Ca²⁺ from intracellular stores. Notably, in the SI villus, chelation of intracellular Ca²⁺ by BAPTA-AM reduced GAP numbers below baseline levels (vehicle: $3.54 \pm 0.34$, BAPTA-AM⁺ CCh, $0.64 \pm 0.10$, $p < 0.001$) demonstrating that both baseline and CCh-induced GAP formation are dependent on intracellular Ca²⁺.

Intracellular stores of Ca²⁺ that are released in response to mAChR activation include the endoplasmatic reticulum (ER) and acidic organelles such as endosomes/lysosomes, and in the case of GCs, mucin granules that represent a large acidic compartment with high Ca²⁺ content (*Wu et al.,*

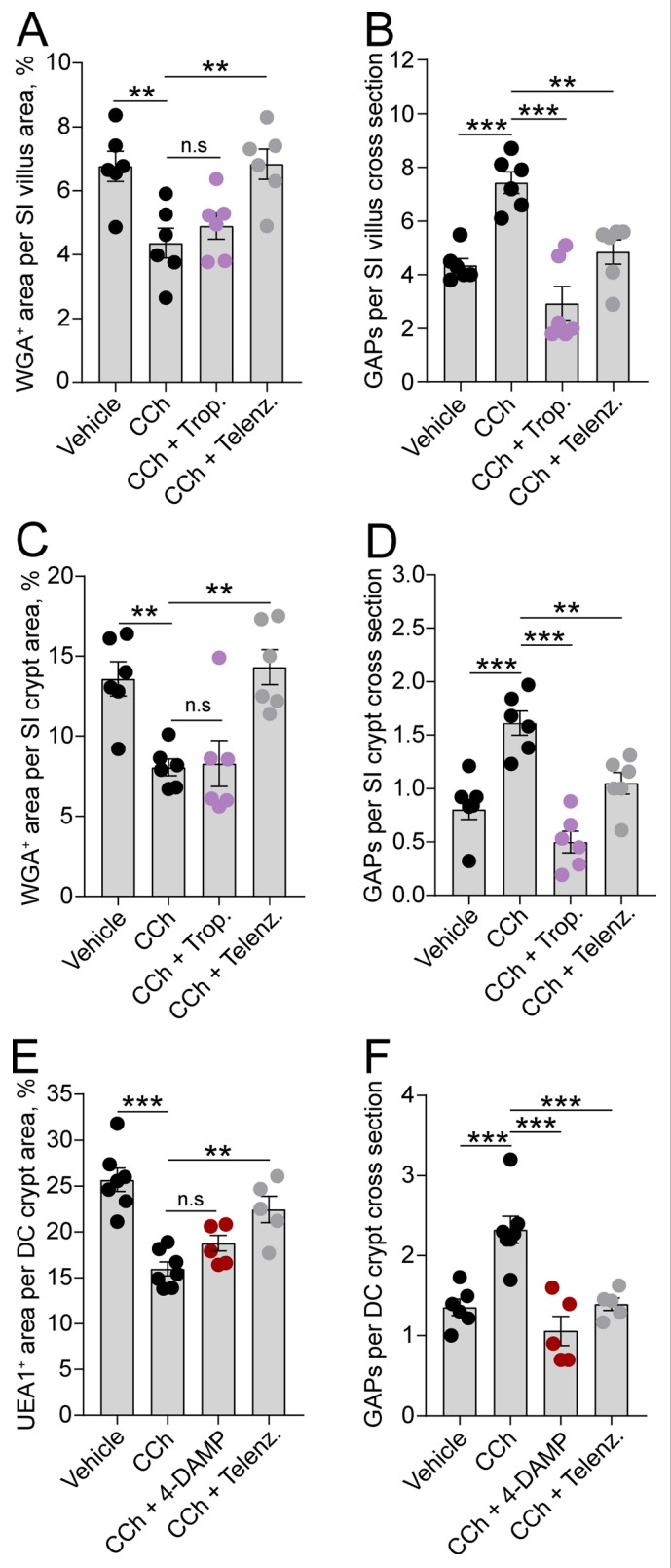

**Figure 5.** Carbamylcholine (CCh)-induced goblet cell-associated antigen passage (GAP) formation and mucus secretion use different muscarinic acetylcholine (ACh) receptors. Effect of the muscarinic ACh receptor 4 (mAChR4) antagonist tropicamide (Trop.), the mAChR1 antagonist telenzepine (Telenz.), or the preferential mAChR3 antagonist 1,1-dimethyl-4-diphenylacetoxypiperidinium iodide (4-DAMP) on CCh-induced mucus secretion and

*Figure 5 continued on next page*

*Figure 5 continued*

GAP formation in the small intestine (SI) villus (**A–B**), the SI crypt (**C–D**), and the distal colon crypt (**E–F**). Data are presented as mean ± SEM. **p < 0.01, ***p < 0.001, n.s = non-significant, as compared to CCh. **A–B**, **D–E**, n = 6 in all groups. (**C and F**) Vehicle and CCh n = 7, CCh + 4 DAMP and CCh + Telenz. n = 5. Statistical analysis was performed using a one-way ANOVA followed by Dunnet's post hoc test. Each data point represents the average of 25 villi or 40 crypts from one mouse.

The online version of this article includes the following figure supplement(s) for figure 5:

**Figure supplement 1.** Small intestine expression of muscarinic ACh receptor 4 (mAChR4) and mAchR1.

**Figure supplement 2.** Distal colon expression of muscarinic ACh receptor 3 (mAChR3) and mAChR1.

---

*2001*). $Ca^{2+}$ release from these intracellular stores occurs via three different pathways mediated by the second messengers inositol trisphosphate (IP3), cyclic ADP ribose (cADPr), and nicotinic acid adenine dinucleotide phosphate (NAADP), each acting on IP3 receptors (IP3R), ryanodine receptors (RyR), and two pore channels (TPC), respectively (*Calcraft et al., 2009*). To explore the role of $Ca^{2+}$ release from intracellular stores in the CCh-induced mucus secretory response and GAP response, we inhibited these signaling pathways; IP3R using xestospongin C (Xesto C), cADPr using 8-Br-cADPr, and NAADP using trans-Ned-19 (T-Ned-19), and evaluated the effects on CCh-induced mucus secretion and GAP formation. In the SI villus, despite the CCh secretory response being reversed by chelation of intracellular $Ca^{2+}$, none of the respective inhibitors reversed the CCh-induced secretory response, suggesting possible redundancy in the signaling pathways inducing this response (*Figure 6A*, F and I-K). The CCh-induced mucus secretion was dependent on NAADP, but independent of IP3R or cADPr in the SI crypts (*Figure 6B*, F and I-K) and distal colon crypts (*Figure 7A*, D and G-I). The GAP response was on the other hand dependent on NAADP and cADPr but independent of IP3R in the SI (*Figure 6C*, D, F and I-K) and distal colon (*Figure 7B*, D and G-I). Since the GAP response was shown to be dependent on intracellular $Ca^{2+}$, we evaluated the location of the ER within GCs forming GAPs. Immunostaining of tissue sections using the ER marker Calnexin showed positive staining throughout the GC, with the exeption of the theca. The ER was observed in close proximity to the dextran at the apical, lateral, and basal sides of the cell (*Figure 6—figure supplement 1*). Thus, the CCh-induced mucus secretory response involves release of intracellular $Ca^{2+}$ via NAADP-mediated pathways and influx of extracellular $Ca^{2+}$, while the GAP response involves release of intracellular $Ca^{2+}$ via NAADP- and cADPr-mediated pathways but is not dependent upon influx of extracellular $Ca^{2+}$. Furthermore, the finding that CCh-induced GAP formation occurred in the absence of CCh-induced mucus secretion in tissues treated with the extracellular $Ca^{2+}$ chelator EGTA, suggests that the role of mAChR1 in driving the CCh-induced increase in GAPs occurs independently of its role in inducing mucus secretion.

## ACh-induced GAP formation and mucus secretion can occur in parallel in the same cells

The findings that CCh-induced GAP formation can occur in the absence of CCh-induced mucus secretion, and that CCh-induced mucus secretion can occur in the absence of GAP formation could either be interpreted as the two processes being induced in separate populations of GCs or that they can occur in parallel in the same GCs but are not functionally linked or dependent upon each other. To explore this question, we evaluated the effect of CCh on the theca area, as a surrogate for mucus secretion, in GCs forming GAPs (epithelial cells staining positive for WGA/UEA1 and tetramethyl-rodamine [TRITC]-dextran) and in those not forming GAPs (epithelial cells staining positive for WGA/UEA1 but negative for TRITC-dextran). We observed that CCh induced a significant decrease in the average theca area of GCs forming GAPs at all three locations (*Figure 8A*), indicating that GCs forming GAPs also respond to CCh with accelerated mucus secretion. The secretory response was also seen as a shift in the size distribution of the theca area in all three locations (*Figure 8B–D*). The CCh-induced decrease in theca area was reversed by EGTA at all three locations (*Figure 8A–D*), showing that GCs forming GAPs respond to CCh with a mucus secretory response mediated by influx of extracellular $Ca^{2+}$. Similar to these findings, GCs in the SI villus and crypt that did not form GAPs in response to CCh responded with a decrease in the average theca area (*Figure 8E*) and a shift in the size distribution of the GCs theca, which was reversed by EGTA (*Figure 8F* and G), consistent with these GCs also responding to CCh with a mucus secretory response mediated by influx of extracellular $Ca^{2+}$. In

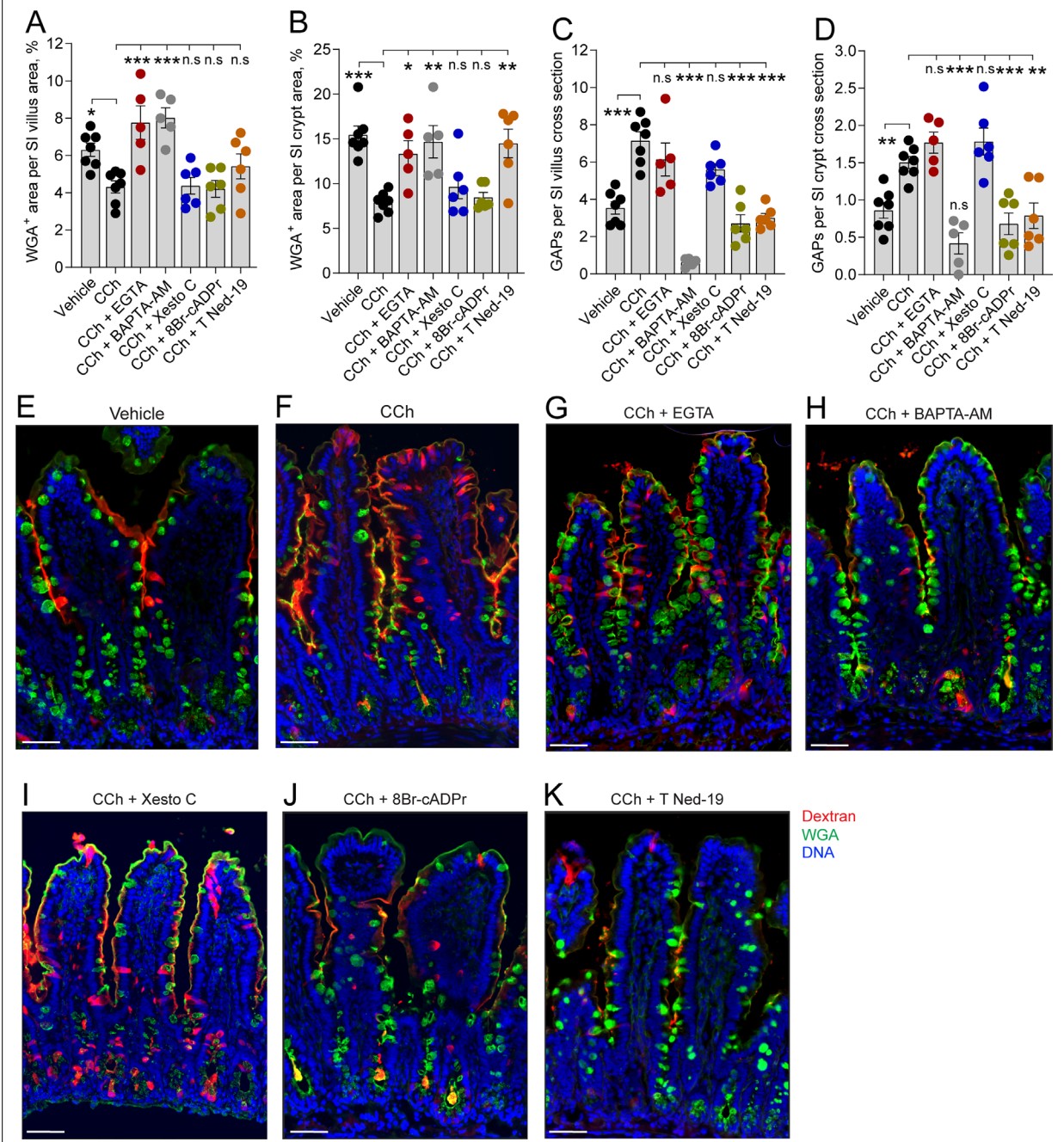

**Figure 6.** In the small intestine (SI), carbamylcholine (CCh)-induced goblet cell-associated antigen passage (GAP) formation and mucus secretion use different calcium pools and signaling pathways. Effect of the extracellular $Ca^{2+}$ chelator EGTA, intracellular $Ca^{2+}$ chelator BAPTA-AM, IP3R inhibitor Xestospongin C (Xesto C), cADPr inhibitor 8-Br-cADPr, and NAADP inhibitor Trans-Ned-19 (T Ned-19) on CCh-induced mucus secretion in (**A**) the SI villus, (**B**) the SI crypt. Effect of $Ca^{2+}$ signaling inhibitors on CCh-induced GAP formation in (**C**) the SI villus, (**D**) the SI crypt. (**E–K**) Representative images of the effect of the respective treatments on CCh-induced mucus secretion and GAP formation. Scale bar: **E–K** = 50 µm. Data are presented as mean ± SEM. *p < 0.05, **p < 0.01, ***p < 0.001, n.s = non-significant as compared to CCh. Vehicle and CCh, n = 7, EGTA and BAPTA-AM n = 5, Xesto C, T-Ned-19, and 8-Br-cADPr n = 6. Statistical analysis was performed using a one-way ANOVA followed by Dunnet's post hoc test. Each data point in (**A–D**) represents the average of 25 villi or 40 crypts from one mouse.

The online version of this article includes the following figure supplement(s) for figure 6:

**Figure supplement 1.** Endoplasmatic reticulum (ER) staining in intestinal goblet cell-associated antigen passage (GAPs).

contrast, distal colon GCs not forming GAPs did not respond to CCh with a reduction in the average theca area or a shift in the theca size distribution, and EGTA had no effect on either average size or the

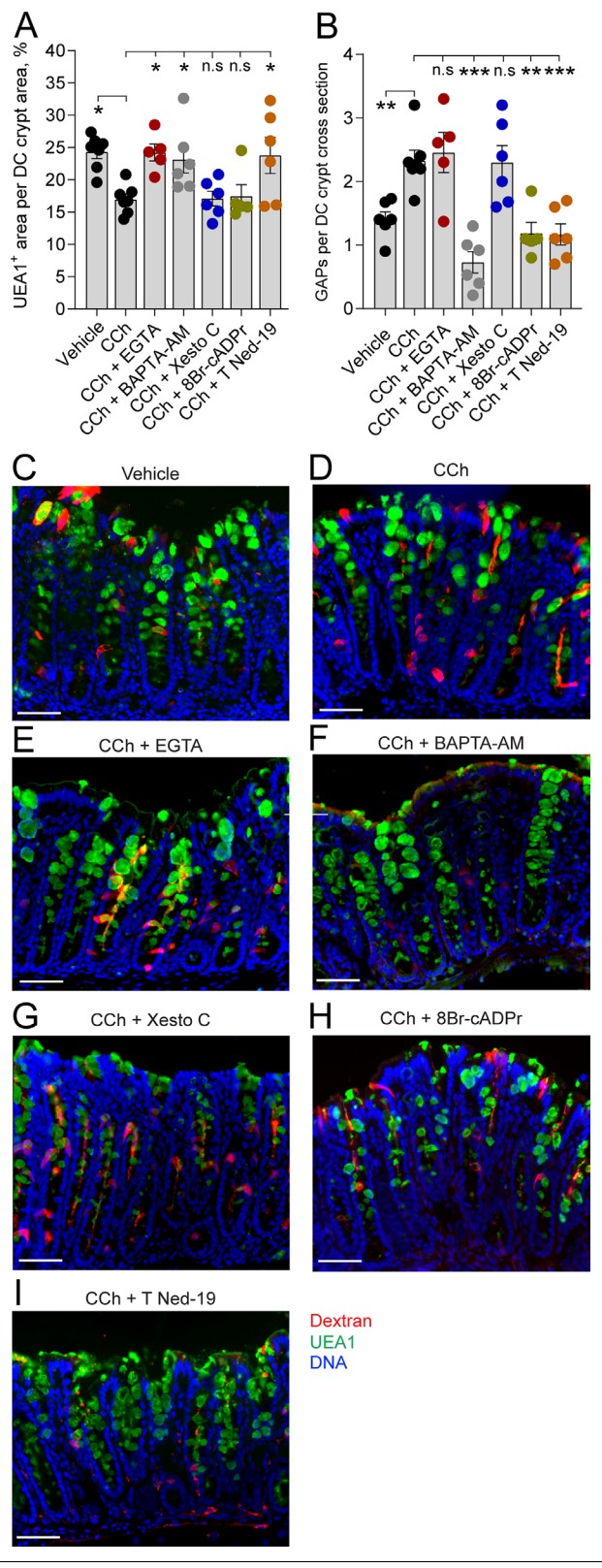

**Figure 7.** In the distal colon, carbamylcholine (CCh)-induced goblet cell-associated antigen passage (GAP) formation and mucus secretion use different calcium pools and signaling pathways. Effect of the extracellular $Ca^{2+}$ chelator EGTA, intracellular $Ca^{2+}$ chelator BAPTA-AM, IP3R inhibitor Xestospongin C (Xesto C), cADPr inhibitor 8-Br-cADPr, and NAADP inhibitor Trans-Ned-19 (T Ned-19) on (**A**) CCh-induced mucus secretion and (**B**) GAP

*Figure 7 continued on next page*

*Figure 7 continued*

formation in the distal colon. (**C–I**) Representative images of the effect of the respective treatments on CCh-induced mucus secretion and GAP formation. Data are presented as mean ± SEM. *p < 0.05, **p < 0.01, ***p < 0.001, n.s = non-significant as compared to CCh. Vehicle and CCh, n = 7, EGTA and 8-Br-cADPr n = 5, BAPTA-AM, Xesto C, and T-Ned-19 n = 6. Scale bar: C–I = 50 µm Statistical analysis was performed using a one-way ANOVA followed by Dunnet's post hoc test. Each data point in (**A–B**) represents the average of 40 crypts from one mouse.

size distribution (*Figure 8E and H*). Given our prior observation that GCs in the distal colon are more likely to completely empty in response to CCh, this suggests that GCs in the distal colon that did not form GAPs, but responded to CCh with accelerated mucus secretion, completely emptied their theca and were lost to our analysis of the theca size.

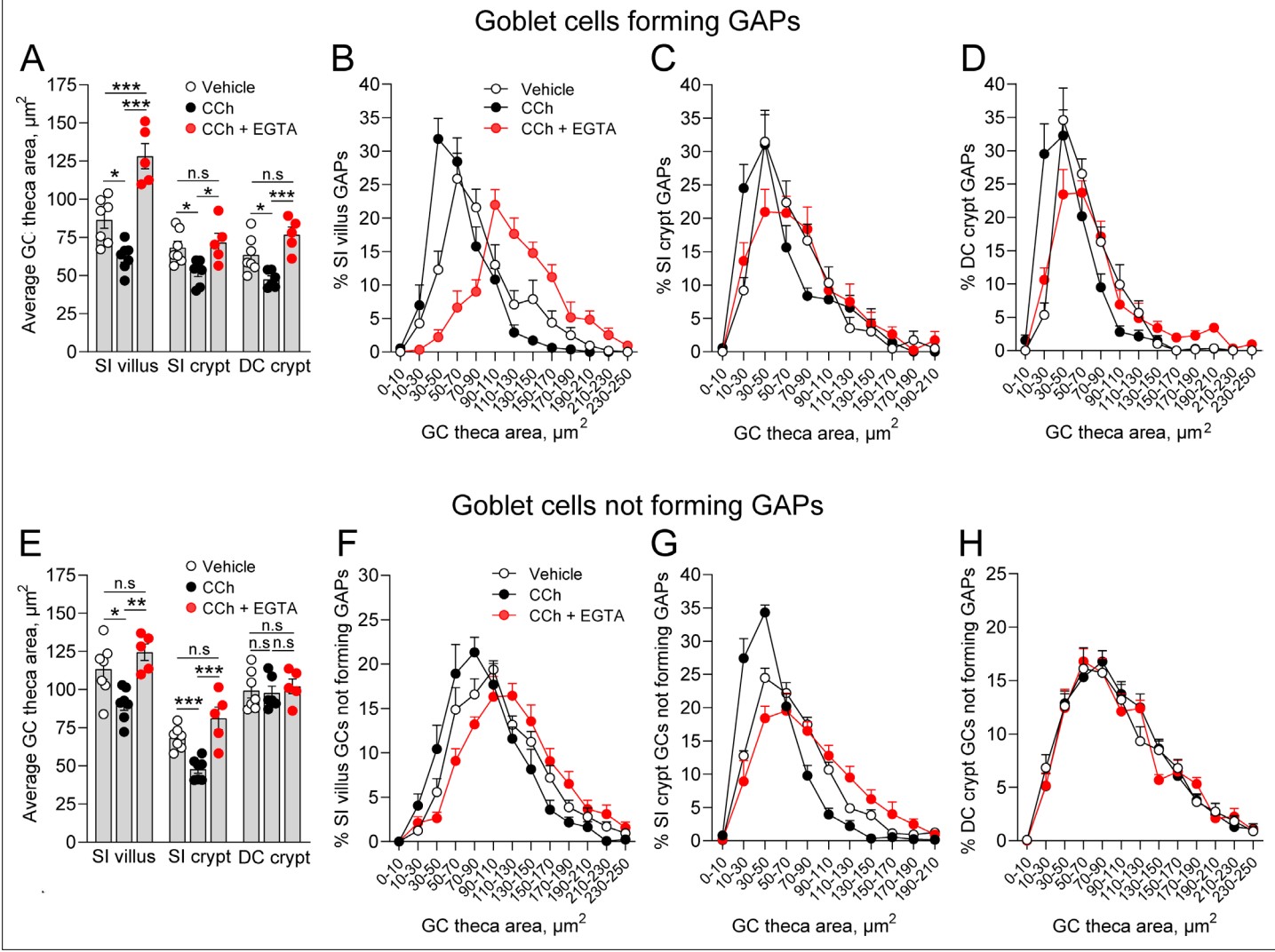

**Figure 8.** Carbamylcholine (CCh) induced mucus secretion and goblet cell (GC)-associated antigen passage (GAP) formation can occur in parallel within the same GC. (**A**) Average theca area of GCs forming GAPs in vehicle, CCh, and CCh + EGTA treated mice. Size distribution of the theca area of GCs forming GAPs in vehicle, CCh, and CCh + EGTA treated mice in (**B**) small intestine (SI) villus, (**C**) SI crypt, and (**D**) DC crypt. (**E**) Average theca area of GCs not forming GAPs in vehicle, CCh, and CCh + EGTA treated mice. Size distribution of GCs not forming GAPs in vehicle, CCh, and CCh + EGTA treated mice in (**F**) SI villus, (**G**) SI crypt, and (**H**) DC crypt. Data are presented as mean ± SEM. Vehicle n = 7, CCh n = 7, CCh + EGTA n = 5. Statistical analysis was performed using a one-way ANOVA with Tukey's post hoc test for multiple comparisons. In panels B–D and F–H, the data represent the percentage of GCs within the respective area bins as part of the total population of GCs forming GAPs, or the total population of GCs not forming GAPs (100%).

The online version of this article includes the following source data for figure 8:

**Source data 1.** Tables of raw data for *Figure 8B–D and F–H*.

Notably, in vehicle treated mice, particularly in the SI villus, the theca area of GCs forming GAPs was generally smaller as compared to GCs not forming GAPs (compare *Figure 8A and E*), consistent with GAP formation occurring in GCs with high baseline mucus secretion. However, the observation that inhibition of CCh-induced mucus secretion did not affect the ability of GCs to form GAPs suggests that the large exocytic event driving CCh-induced mucus secretion is different from the exocytic event preceding GAP formation. In aggregate, these results demonstrate that activation of select muscarinic receptor subtypes and their respective downstream signaling pathways allows GCs to respond to ACh with either GAP formation and antigen uptake, mucus secretion, or with both processes in parallel.

## Discussion

The intestinal epithelium is faced with the complex task of acting as a semi-permeable barrier allowing efficient nutrient absorption and controlled exposure of the immune system to lumenal antigens to sustain immune tolerance, while at the same time limiting contact with harmful agents that pass through the gastrointestinal tract. GCs have long been appreciated for their role in barrier function via production and secretion of the mucus layer that covers the intestinal surfaces (*Johansson et al., 2008*; *Van der Sluis et al., 2006*). Recently, the role of GCs in intestinal homeostasis has expanded to include participation in adaptive immune responses to lumenal substances via sampling and delivery of lumenal antigens to lamina propria antigen presenting cells (*Knoop et al., 2015*; *McDole et al., 2012*). How GCs balance these seemingly opposing tasks and the cellular mechanisms underlying lumenal antigen sampling were largely unknown. In the present study, we demonstrate that GCs acquire lumenal substances via an endocytic event that efficiently delivers fluid-phase cargo not only to lysosomes, MVBs, and the TGN, but also into the transcytotic pathway allowing the capture of lumenal substances by underlying cells (*Figure 9*). Such transcellular transport of lumenal solutes was not observed in adjacent enterocytes, and appears to be a feature specific to intestinal secretory cells and potentially linked to retrieval of secretory granule membrane. We cannot exclude that transcellular transport of luminal substances captured by fluid-phase endocytosis in enterocytes occurs to some degree, but if so, the efficiency of such transport must be low as expected for fluid-phase endocytic cargo in enterocytes, and below the level of detection in our assays.

Endocytic retrieval of secretory granule membranes has been shown to occur in various secretory cells including neurons, enterochromaffin cells, pancreatic acinar cells, and endothelial cells (*Henkel et al., 2001*; *Stevenson et al., 2017*; *Wen et al., 2012*) and is linked to membrane recycling and regulation of the secretory capacity of the cell (*Stevenson et al., 2017*; *Wen et al., 2012*). The general consensus regarding exocytosis – endocytosis coupling in secretory cells is that during primary exocytosis, the process where one granule at the time is inserted into the plasma membrane, inserted membranes have to be retrieved to enable subsequent rounds of exocytosis and to restore plasma membrane structure. However, during compound exocytosis, the process where multiple secretory granules fuse with one another and release their content through a common secretion pore, large volumes of membrane retrieval are not necessarily required to sustain secretion as individual secretory granules are not inserted directly into the plasma membrane (*Liang et al., 2017*). Our previous studies demonstrate that not all GCs form GAPs (*Knoop et al., 2015*; *McDole et al., 2012*), that GAP formation is not required to maintain the mucus barrier (*Kulkarni et al., 2020*), and our current findings that blockade of GAP formation does not inhibit mucus secretion, indicate that GAP formation cannot be required to maintain mucus secretion or the mucus barrier. Though compensatory endocytosis must exist following secretion in GCs, to balance membrane trafficking, the fate of these endocytic vesicles does not necessarily intersect with the GAP pathway. Rather, we suggest that GAPs represent a specialized endocytic pathway enabling enhanced and regulated transcellular transport of gut lumenal solutes across the epithelial barrier by transcytosis. We consider it likely the GAP pathway evolved from adaptations to the secretory and membrane recovery machinery specific to this highly secretory cell type.

Despite the associations between GAP formation and mucus secretion, our results demonstrate that ACh-induced GAP formation and mucus secretion are not functionally linked or dependent upon one another. Rather we found that they are regulated by separate muscarinic receptors and intracellular signaling pathways – allowing them to be performed independently or in parallel in the same cell. These observations provide further evidence that the exocytic events driving ACh-induced mucus secretion can be different from that preceding GAP formation. In spinal cord neurons, exposure to high

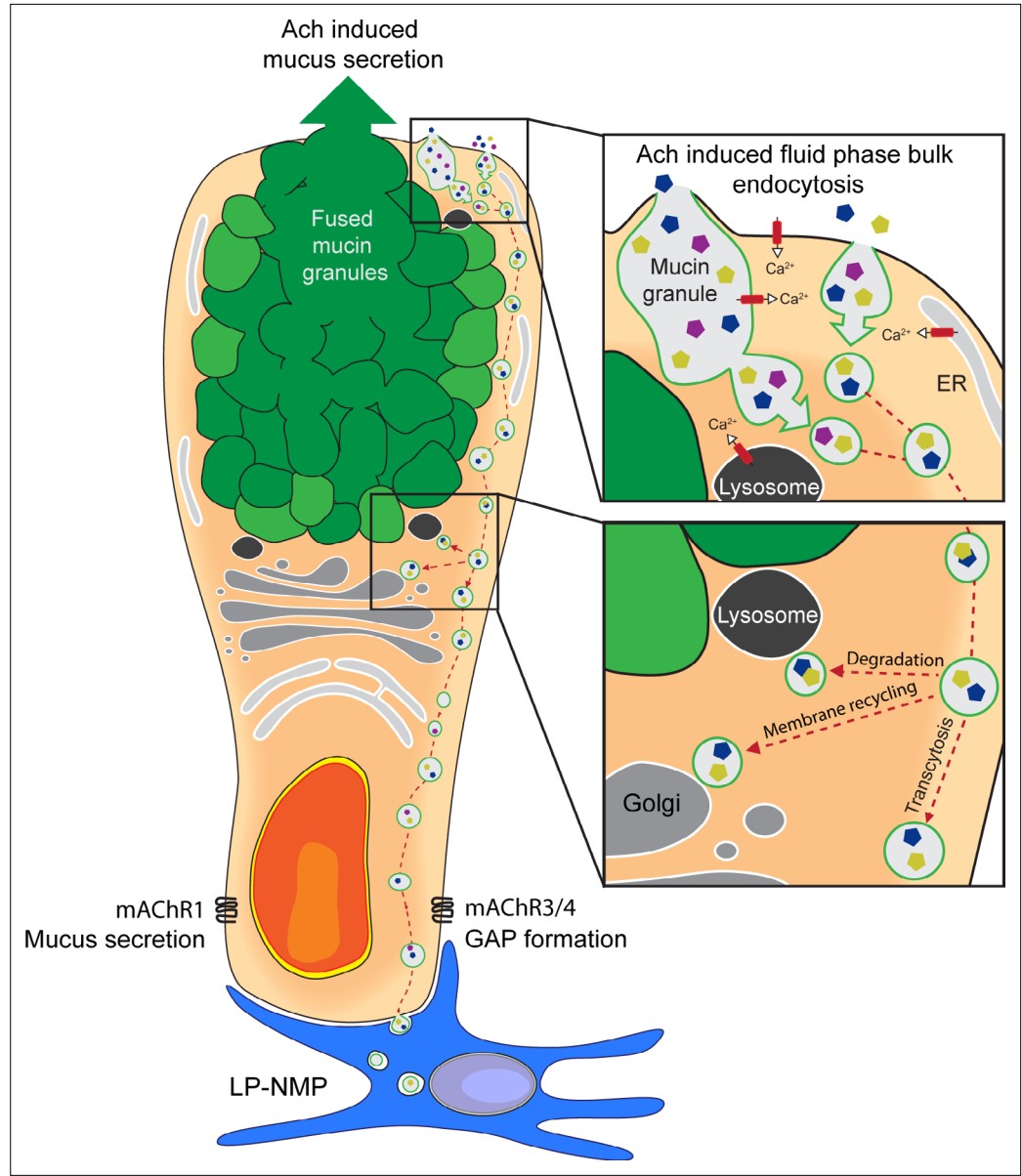

**Figure 9.** Schematic representation of acetylcholine (ACh)-induced mucus secretion and goblet cell-associated antigen passage (GAP) formation in intestinal goblet cells. In response to exogenous ACh, intestinal goblet cells respond with an accelerated mucus secretory response mediated by muscarinic ACh receptor 1 (mAChR1), and/or induction of fluid-phase bulk endocytosis of secretory granule membrane (GAP formation) mediated by mAChR4 in the small intestine (SI) and mAChR3 in the distal colon. The endocytic vesicles containing luminal fluid-phase cargo are shuttled through the cell for degradation, membrane recycling, and transcytosis to be acquired by lamina proporia mononuclear phagocytes (LP-MNPs). Using separate mAChRs, $Ca^{2+}$ pools and signaling pathways for the processes of ACh-induced GAP formation and ACh-induced mucus secretion allow these processes to occur independently or in parallel within the same goblet cell forming the basis of how goblet cells can differentially regulate when to maintain the mucus barrier and when to deliver luminal substances to the immune system.

concentrations of $K^+$ triggers neurotransmitter release paralleled by endocytic retrieval of secretory granule membrane and uptake of high molecular weight substances from the extracellular environment. Similar to our observations demonstrating that inhibition of CCh-induced mucus secretion did not affect CCh-induced GAP formation, inhibition of $K^+$ evoked exocytosis did not affect $K^+$-induced endocytic retrieval of secretory granule membrane. The intact endocytic response in response to elevated $K^+$ levels was shown to be mediated by retrieval of previously fused secretory vesicles, thus,

although elevated $K^+$ triggered secretory granule exocytosis, retrieval of previously fused granules was enough to sustain the uptake process (*Neale et al., 1999*). Applied to our data this would suggest that in situations when ACh-induced mucus secretion does not occur; an intact GAP response can be maintained via retrieval of previously fused secretory granules inserted into the plasma membrane during a previous exocytotic event. In addition to secreting mucus in response to secretagogues such as ACh, GCs secrete mucus constitutively resulting in continuous insertion of secretory granule membrane into the apical membrane to be retrieved in response to ACh (*Oliver and Specian, 1990*). Further support in favor of the two processes being functionally separate, ACh-induced mucus secretion is initiated at the center of the GC theca (*Specian and Neutra, 1980*), while GAPs initiate along the lateral apical surface of the cell and progress basally. This spatial separation of the two processes may imply that GAPs primarily form by retrieval of secretory granules released during steady state, which in contrast to ACh-induced exocytosis are supplied by granules transported along the outer borders of the theca and released at the apical lateral sides (*Oliver and Specian, 1990*). Alternatively, ACh may induce multiple endocytic events, some originating from retrieval of secretory granule membrane, and others originating from endocytosis of the plasma membrane.

Despite our findings of muscarinic receptor subtype-specific regulation of ACh-induced GAP formation and mucus secretion, muscarinic receptor antagonists are known for their receptor subtype promiscuity, and difficulties in assessing differences in in vivo metabolism of antagonists warrant caution when assessing the relative contribution of the respective receptor subtypes (*Bozkurt and Sahin-Erdemli, 2009*; *Erosa-Rivero et al., 2014*; *Lazareno and Birdsall, 1993*). Nonetheless, in support of these results, we also find that the CCh-induced GAP response and the mucus secretory response use different pools of $Ca^{2+}$ and different second messenger systems. Furthermore, the fact that the two processes can be performed separately supports a model of regulation by separate mechanisms. In lacrimal gland acinar cells, exposure to low concentrations of ACh was shown to increase fluid-phase endocytosis by 80 % while protein release was increased only by 40 %. In contrast, exposure to high concentrations of ACh was shown to increase protein release by 80%, while endocytic retrieval was increased by 40 % (*Gierow et al., 1995*), demonstrating that, similar to what we observe in intestinal GCs, in lacrimal acinar cells, ACh-induced exocytosis and endocytosis can, at least to some degree, occur separately. The intracellular $Ca^{2+}$ concentration needed to trigger endocytosis has been shown to be lower than that needed to trigger exocytosis (*Marks and McMahon, 1998*). This may explain why in the steady state, ACh acting on mAChR4 in the SI and mAChR3 in the distal colon induces GAP formation (endocytosis) without an apparent effect on mucus secretion (exocytosis); and yet in response to exogenous administration of high concentrations of ACh, which could trigger a larger increase in intracellular $Ca^{2+}$, GCs respond with both mucus secretion and GAP formation.

When evaluating the frequency of GAPs along the SI crypt villus axis, we observed that villus GCs formed GAPs at a higher frequency when compared to crypt GCs both during steady state and in response to CCh. Crypt GCs on the other hand responded to CCh with a stronger mucus secretory response resulting in complete emptying of a subset of GCs and partial emptying of the remaining GCs, which is in accordance with previously published data (*Phillips, 1992*). These results indicate that in the SI, in response to ACh, villus GCs preferentially perform antigen uptake and delivery to the immune system, while crypt GCs have the capacity to secrete large volumes of mucus when needed. Baseline mucus secretion has been shown to be higher in the SI villus as compared to the crypts which may explain why baseline GAP formation is more prevalent in SI villus GCs as there would be more membrane to retrieve to form GAPs (*Schneider et al., 2018*).

In GCs, mAChR4 signaling (the pathway regulating GAP formation in the SI and proximal colon) is inhibited by activation of the epidermal growth factor receptor (EGFR), which in the proximal colon can occur by GC intrinsic sensing of gut microbial products via Toll-like receptors or by the presence of EGFR ligands in the SI and colon lumen (*Knoop et al., 2015*). The benefit of mucus secretion not being functionally linked to GAP formation is apparent in the proximal colon where both steady-state and CCh-induced GAP formation are inhibited, while baseline and CCh-induced mucus secretion remain intact (*Ermund et al., 2013*; *Knoop et al., 2015*). Overriding the inhibition of mAChR4 signaling to allow GAPs to form in the proximal colon has the undesired outcome of bacterial translocation across GAPs and induction of inflammatory responses (*Knoop et al., 2016*). A further example of the importance of controlling GAP formation is that during enteric infection with *Salmonella typhimurium*, mAChR4 signaling and GAP formation in the SI is inhibited to prevent lumenal antigen

delivery to the immune system and limit inflammatory T cell responses to dietary antigens (*Kulkarni et al., 2018*). Thus, functional uncoupling of ACh induced mucus secretion from GAP formation, the usage of different mAChRs to perform mucus secretion and GAP formation, and the ability to regulate mAChR4 signaling in GCs, underlies the ability to control when and where GAPs are formed and are essential in maintaining intestinal health.

Although the focus of the present study was to explore the cellular basis of ACh-induced GAP formation, the ability of GCs to acquire lumenal substances is not restricted to activation of muscarinic receptor signaling and additional ligands and receptors activating cADPr production, such as IL13, have been demonstrated to induce GCs to take up lumenal substances in some settings (*Noah et al., 2019*). This raises the possibility that other ligands, whose receptors are expressed by GCs and induce cADPr production, may likewise induce GCs to acquire lumenal substances (*Deshpande et al., 2004*; *Tliba et al., 2004*). While it is becoming apparent that physiologic ACh-induced GAP formation supports antigen-specific T cell responses and tolerance to the lumenal content, the functional consequences and downstream events resulting from GC uptake of lumenal substances induced by other stimuli remain to be explored.

In summary, our results define the basis by which GCs sample lumenal antigens for delivery to the immune system and how this process is balanced with mucus secretion within the same cell. Our observations indicate that GCs have evolved a pathway of fluid-phase endocytosis that efficiently serves the transcytotic pathway for non-specific uptake of lumenal substances that are sampled by sub-epithelial phagocytic cells. Compared to neighboring enterocytes, the fluid-phase endocytic GAP pathway of GCs is remarkable for its efficient transport of lumenal substances across the epithelial barrier. Perhaps GAPs evolved from an endocytic system originally used in secretory cell types to recycle secretory granule membranes, but adapted to ensure that during situations when the lumenal environment is toxic for antigen sampling, the GC can suppress GAP formation while retaining its ability to secrete mucus for maintenance of the mucus barrier. In total, our observations reveal the basis by which GCs perform the function of lumenal antigen delivery to the immune system, and provide mechanistic insights into how the critical roles of barrier maintenance and antigen delivery are achieved within the same cell.

# Materials and methods

## Key resources table

| Reagent type (species) or resource | Designation | Source or reference | Identifiers | Additional information |
|---|---|---|---|---|
| Antibody | EEA1 (rabbit monoclonal) | Cell Signaling Technology | Cat#3288 S, RRID:AB_2096811 | IF (1:100) |
| Antibody | Lysozyme (rabbit polyclonal) | Thermo Fisher Scientific | Cat#PA5-16668, RRID:AB_10984852 | IF (1:100) |
| Antibody | Rab3D (rabbit polyclonal) | Synaptic systems | Cat#107 303, RRID:AB_2253547 | IF (1:100) |
| Antibody | Rab7 (rabbit monoclonal) | Cell Signaling Technology | Cat#9367T, RRID:AB_1904103 | IF (1:100) |
| Antibody | LAMP-1 (rabbit polyclonal) | Abcam | Cat#ab62562 RRID:AB_2134489 | IF (1:100) |
| Antibody | TGN46 (rabbit monoclonal) | Thermo Fisher | Cat#MA5-32532 RRID:AB_2809809 | IF (1:100) |
| Antibody | VAMP8 (chicken polyclonal) | University of Texas | Prof Burton Dickey | IF (1:500) |
| Antibody | Calnexin (rabbit polyclonal) | Abcam | Cat#ab22595 RRID:AB_2069006 | IF (1:100) |
| Antibody | Goat anti-Chicken IgY Alexa Fluor 555 (goat polyclonal) | Thermo Fisher Scientific | Cat#A32932 RRID:AB_2762844 | IF (1:1000) |
| Antibody | Goat anti-Rabbit IgG Alexa Fluor 488 (goat polyclonal) | Thermo Fisher Scientific | Cat#A-11008, RRID:AB_143165 | IF (1:1000) |

*Continued on next page*

*Continued*

| Reagent type (species) or resource | Designation | Source or reference | Identifiers | Additional information |
|---|---|---|---|---|
| Antibody | Goat anti-Rabbit IgG Alexa Fluor 647 (goat polyclonal) | Thermo Fisher Scientific | Cat#A-21244 RRID:AB_2535812 | IF (1:1000) |
| Chemical compound, drug | Atropine | Sigma-Aldrich | Cat#A0257 | 550 µg/kg i.p. |
| Chemical compound, drug | Telenzepine | Sigma-Aldrich | Cat#T122 | 550 µg/kg i.p. |
| Chemical compound, drug | 4-DAMP | Sigma-Aldrich | Cat#SML0255 | 550 µg/kg i.p. |
| Chemical compound, drug | Tropicamide | Tocris | Cat#0909 | 100 µg/kg s.c. |
| Chemical compound, drug | Carbamylcholine | Sigma-Aldrich | Cat#C4382 | 125 µg/kg s.c. |
| Chemical compound, drug | FM 1–43 FX | Thermo Fisher Scientific | Cat#F35355 | 50 µg/ml |
| Chemical compound, drug | EGTA | Sigma-Aldrich | Cat#E3889 | 2 mM |
| Chemical compound, drug | BAPTA-AM | Sigma-Aldrich | Cat#A1076 | 200 µM |
| Chemical compound, drug | Xestospongin C | Sigma-Aldrich | Cat#X2628 | 22 µM |
| Chemical compound, drug | Dynasore | Sigma-Aldrich | Cat#D7693 | 150 µM |
| Chemical compound, drug | Dyngo 4a | Selleck Chemicals | Cat#S7163 | 120 µM |
| Chemical compound, drug | Colchicine | Tocris | Cat#1364 | 100 µM |
| Chemical compound, drug | Cytochalasin D | Tocris | Cat#1233 | 4 µM |
| Chemical compound, drug | Ciliobrevin D | Sigma-Aldrich | Cat#250401 | 100 µM |
| Chemical compound, drug | Dimethylenastron (DMEA) | Tocris | Cat#5261 | 10 µM |
| Chemical compound, drug | 8-Br-cADPr | Sigma-Aldrich | Cat#B5416 | 0.2 mg/kg i.p. |
| Chemical compound, drug | Trans-Ned19 | Tocris | Cat#3954 | 5 mg/kg i.p |
| Chemical compound, drug | LY294002 | Sigma-Aldrich | Cat#L9908 | 4 mg/kg i.p |
| Chemical compound, drug | Diamidino-2-phenylindole (DAPI) | Sigma-Aldrich | Cat#D9542 | 1 µg/ml |
| Chemical compound, drug | UEA1 Fluorescein | Vector Laboratories | Cat#FL-1061–2 | 10 µg/ml |
| Chemical compound, drug | WGA Fluorescein | Vector Laboratories | Cat#FL-1021 | 10 µg/ml |
| Chemical compound, drug | WGA Texas Red | Sigma-Aldrich | Cat#W21405 | 10 µg/ml |
| Chemical compound, drug | Dextran tetramethylrodamine conjugate, lysine fixable, MW 10,000 | Thermo Fisher Scientific | Cat#D1817 | 12.5 mg/ml |

*Continued*

| Reagent type (species) or resource | Designation | Source or reference | Identifiers | Additional information |
|---|---|---|---|---|
| Chemical compound, drug | Dextran biotin conjugate, lysine fixable, MW 10,000 | Thermo Fisher Scientific | Cat#D1956 | 12.5 mg/ml |
| Chemical compound, drug | Dextran Alexa 647 conjugate lysine fixable, MW 10,000 | Thermo Fisher Scientific | Cat#D22914 | 12.5 mg/ml |
| Chemical compound, drug | Ovalbumin Texas Red conjugate | Thermo Fisher Scientific | Cat#O23021 | 12.5 mg/ml |
| Chemical compound, drug | Ni DAB | Vector Laboratories | Cat#SK-4100, RRID:AB_2336382 | |
| Commercial assay or kit | ABC Elite kit | Vector Laboratories | Cat#PK-6100, RRID:AB_2336819 | |
| Commercial assay or kit | RNAscope fluorescent reagent kit v2 | ACD, Bio-Techne | Cat# 323100 | |
| Sequence-based reagent | Mm-Chrm1-O1-C2 | ACD, Bio-Techne | Cat#483441-C2 | |
| Sequence-based reagent | Mm-Chrm3 | ACD, Bio-Techne | Cat#437701 | |
| Sequence-based reagent | Mm-Chrm4-C2Chrm4 | ACD, Bio-Techne | Cat#410581-C2 | |
| Species, species background (*Mus musculus*) | Mouse: C57 Bl/6 J | Jackson Laboratories | Stock No: 000664, RRID:IMSR_JAX:000664 | |
| Species, species background (*Mus musculus*) | Mouse: Itgax YFP | Jackson Laboratories | Stock No: 008829 RRID: IMSR_JAX:008829 | |
| Species, species background (*Mus musculus*) | Mouse: Cx3cr1 GFP | Jackson Laboratories | Stock No: 005582 RRID: IMSR_JAX:005582 | |
| Software, algorithm | GraphPad Prism 7 | GraphPad | https://www.graphpad.com, RRID:SCR_002798 | |
| Software, algorithm | Amira | Thermofisher | http://www.fei.com/software/amira-3d-for-life-sciences/, RRID:SCR_007353 | |
| Software, algorithm | Zen | Zeiss | http://www.zeiss.com/microscopy/en_us/products/microscope-software/zen.html#introduction, RRID:SCR_013672 | |
| Software, algorithm | Fiji | *Schindelin et al., 2012* | https://fiji.sc/, RRID:SCR_002285 | |
| Software, algorithm | AxioVision | Zeiss | http://www.zeiss.com/microscopy/en_de/products/microscope-software/axiovision-for-biology.html, RRID:SCR_002677 | |
| Software, algorithm | Imaris | Bitplane | http://www.bitplane.com/imaris/imaris RRID:SCR_007370 | |
| Software, algorithm | ImageWarp | A&B Software | http://www.imagewarp.com/ | |

## Experimental model and subject details

### Mice

All mice were 10 or more generations on the C57BL/6 background. C57BL/6 mice, *Itgax^YFP* transgenic mice (*Lindquist et al., 2004*), and *Cx3cr1^GFP* transgenic mice (*Jung et al., 2000*) were purchased from The Jackson Laboratory (Bar Harbor, ME) and bred in house. Mice were housed in a specific pathogen-free facility and fed a routine chow diet. Mice of both sexes were used in this study as no gender-specific differences have been observed in previous studies. The age of the mice ranged between 7 and 12 weeks, and animals were randomly assigned to the respective experimental groups. All animal procedures and protocols were performed in accordance with the Institutional Animal Care and Use Committee at Washington University School of Medicine (Animal Welfare Assurance number: A-3381–01) and the Swedish animal welfare legislation and approved by the Swedish Laboratory Animal Ethical Committee in Gothenburg (Ethical permit ID number: 5.8.18-11053/2019).

## Method details

### Reagents and treatments

The muscarinic receptor antagonists atropine, telenzepine, and 4-DAMP (Sigma-Aldrich, St Louis, MO) were administered i.p. at a dose of 550 µg/kg, 30 min prior to intraluminal injection of 10 kDa lysine fixable TRITC-dextran (12.5 mg/ml) (Thermofisher, Waltham, MA). Tropicamide (Tocris, Minneapolis, MN) was administered at a dose of 100 µg/kg s.c. 20 min prior to intraluminal injection of TRITC-dextran, and every 20 min during the 60 min incubation period. CCh (Sigma-Aldrich, St Louis, MO) was administered s.c. at a dose of 125 µg/kg, 5 min following intraluminal injection of TRITC-dextran. FM 1–43 FX (Thermo Fisher Scientific, Waltham, MA) was administered intraluminal at a dose of 50 µg/ml, followed by 1 hr incubation. EGTA (2 mM), BAPTA-AM (200 µM), xestospongin C (22 µM), dynasore (150 µM), ciliobrevin D (100 µM) (Sigma-Aldrich, St Louis, MO), Dyngo 4a (120 µM) (Selleck Chemicals, Houston, TX), colchicine (100 µM), cytochalasin D (4 µM), and dimethylenastron (DMEA) (10 µM) (Tocris, Minneapolis, MN) were administered intraluminal 30 min prior to intraluminal injection of TRITC-dextran. In these experiments, the TRITC-dextran solution also contained the respective inhibitors at the concentration described above. 8-Br-cADPr (0.2 mg/kg) (Sigma-Aldrich, St Louis, MO) and Trans-Ned19 (5 mg/kg) (Tocris, Minneapolis, MN) were administered i.p. 30 min prior to intraluminal injection of TRITC-dextran and LY294002 (4 mg/kg) was administered i.p. 15 min prior to intraluminal injection of TRITC-dextran.

### Intravital two-photon microscopy

Mice were anesthetized using nebulized isofluorane in 95 % $O_2$/5 % $CO_2$. Intravital preparation of the SI was performed as previously described (*McDole et al., 2012*). Briefly, a vertical midline incision was made through the skin and peritoneum revealing the SI. To label GAPs, a mixture of 10 kDa TRITC-dextran (10 mg/ml), and 4',6-diamidino-2-phenylindole (DAPI) (10 mg/ml) (Thermo Fisher Scientific, Waltham, MA) was injected intraluminal into the jejunum, followed by 20 min incubation. To image the luminal surface, a 10 mm incision was made to the dextran-containing intestinal segment. Large fecal particles were removed with care using a blunt forcep, and smaller particles were left in place to avoid disrupting the mucus layer and the epithelium. Imaging was performed with the intestine remaining within the peritoneal cavity. The tissue was excited using a Chameleon XR Ti: sapphire laser tuned to 890 nm (Coherent, Santa Clara, CA). Images were acquired using a custom-built 2 P microscope running ImageWarp software (A&B Software, New London, CT). Acquired images were analyzed using Imaris (Bitplane, Belfast, Great Britain).

### Enumeration of GAPs

Lysine fixable TRITC or Alexa Fluor 647 conjugated 10 kDa dextran was administered into the SI or distal colon of ketamine/xylasine anesthetized mice. After 1 hr, mice were sacrificed, tissues were dissected, cut open along the mesenteric border, and thoroughly washed with cold PBS before fixation in 10 % formalin buffered solution for 2 hr, followed by incubation in 30 % sucrose overnight. Tissues were embedded in optimal cutting temperature compound (OCT) (Fisher Scientific, Pittsburgh, PA) and 6 µm sections were prepared. Sections were stained with fluorescein conjugated WGA (Vector Laboratories, Burlingame, CA) or fluorescein conjugated UEA1 (Vector Laboratories, Burlingame, CA) to identify GCs in the SI and distal colon, respectively, and stained with DAPI (Sigma-Aldrich, St Louis, MO) to identify the nucleus. Sections were imaged using an Axioskop 2 microscope (Carl Zeiss Microscopy, Thornwood, NY) or a Nikon n-SIM super-resolution microscope (Nikon Instruments, Melville, NY). Obtained images were evaluated using AxioVision, and Imaris, respectively. GAPs were identified as TRITC-dextran filled WGA+ (SI)/UEA1+ (distal colon) epithelial cells measuring approximately 20 µm (height) × 5 µm (diameter) traversing the epithelium and containing a nucleus and were enumerated as GAPs per villus/crypt cross section in the SI, and GAPs per crypt cross section in the distal colon.

### Quantification of mucus secretion

Mucus secretion was quantified using Fiji (*Schindelin et al., 2012*) by measuring the area of the WGA+ (SI) or UEA1+ (distal colon) GCs theca, and expressed either as GC theca area in µm² or as the total mucus content per villus/crypt cross section, quantified as the sum of all WGA+/UEA1+ GC theca areas, divided by the area of the villus/crypt cross section, expressed as percentage. Mucus secretion

was defined as a significant decrease in the GC theca area, or total mucus content of the villus/crypt cross section comparing vehicle control and the respective treatment groups.

## Evaluation of GC endocytosis using FM 1–43FX

Briefly, FM 1–43 FX was administered into the SI of ketamine/xylasine anesthetized mice. After 1 hr, mice were sacrificed and the SI was dissected, cut open along the mesenteric border, and washed with cold PBS before fixation in 10 % formalin buffered solution for 2 hr, followed by incubation in 30 % sucrose overnight. Tissues were embedded in OCT, and 6 μm sections were prepared. To evaluate the FM 1–43 FX staining pattern in GC, sections were stained with Texas Red conjugated WGA (Thermo Fisher Scientific, Waltham, MA), and DAPI to identify the nucleus, and imaged using an Axioskop 2 microscope. Images were analyzed using AxioVision (Carl Zeiss Microscopy, Thornwood, NY). To evaluate the FM 1–43 FX staining pattern in GCs forming GAPs, FM 1–43 FX and dextran-Alexa647 was administered into the SI lumen simultaneously. After 1 hr the mice were sacrificed and tissue processes as described above. Tissue sections were stained with DAPI to identify the nucleus and imaged using an LSM700 Axio Examiner Z1 confocal microscope (Carl Zeiss Microscopy, Thornwood, NY). Acquired images were analyzed using Zen (Carl Zeiss Microscopy, Thornwood, NY) and Imaris (Bitplane, Belfast, Great Britain) software.

## Immunohistochemistry

Tissue specimens were washed in cold PBS, fixed in 10% formalin buffered saline for 2 hr, followed by incubation in 30% sucrose overnight. Tissue specimens were embedded in optimal tissue cutting medium and 6 -μm-thick section were cut. Sections were rinsed in PBS followed by antigen retrieval in 0.05% citraconic anhydride (Sigma-Aldrich, St Louis, MO), permeabilized in 0.05% Triton X-100 or 0.05% saponin (TGN46) (Sigma-Aldrich, St Louis, MO), washed in PBS, and blocked in 5% goat serum (Sigma-Aldrich, St Louis, MO). Sections were incubated with primary antibodies toward EEA1, Rab7 (Cell Signaling Technology, Beverly, MA), Lysozyme, TGN46 (Thermo Fisher Scientific, Waltham, MA), Rab3D (Synaptic Systems, Goettingen, Germany), LAMP1, Calnexin (Abcam, Cambridge, United Kingdom), or VAMP-8 (gift from Prof Burton Dickey, University of Texas) at 4°C overnight, rinsed with PBS and incubated with goat anti rabbit or goat anti chicken Alexa Fluor conjugated secondary antibodies (Thermo Fisher Scientific, Waltham, MA) at room temperature for 1 hr. Sections were counterstained with DAPI and imaged using an Axioskop 2 microscope, an LSM700 Axio Examiner Z1 confocal microscope, or an LSM900 with Airyscan 2 microscope (Carl Zeiss Microscopy, Thornwood, NY). Acquired images were analyzed using Zen (Carl Zeiss Microscopy, Thornwood, NY) and Imaris (Bitplane, Belfast, Great Britain) software.

## RNAscope

SI and distal colon tissue specimens were dissected, fixed for 16 hr in 4% formaldehyde, followed by overnight incubation in 30% sucrose. Tissue specimens were embedded in optimal cutting medium, frozen, and 6 μm thick sections were cut. Cut sections were washed in PBS, followed by single molecule in situ hybridization of Chrm1 (probe: 483441-C2), Chrm3 (probe: 437701), or Chrm4 (probe: 410581-C2) according to manufacturer's instructions RNAscope fluorescent reagent kit v2 (ACD, Bio-Techne, Abingdon, United Kingdom) visualized with Opal-570 (Perkin-Elmer, Waltham, MA) or CF488 A Tyramine (Biotium, Fremont, CA). Following the RNAscope protocol, tissue sections were stained with a primary antibody against Epcam (Abcam, Cambridge, United Kingdom) at 4°C overnight followed by incubation using a Alexa Fluor 647 conjugated secondary antibody (Thermo Fisher Scientific, Waltham, MA). Nuclei were counterstained with Hoechst 33528 and a subset of SI sections were stained with Alexa Fluor 555 conjugated WGA (Thermo Fisher Scientific, Waltham, MA), and a subset of distal colon sections were stained with FITC conjugated UEA1 (Vector Laboratories, Burlingame, CA) to visualize GCs. Images were obtained by an LSM700 Axio Examiner Z1 confocal microscope (Carl Zeiss Microscopy, Thornwood, NY).

## FIB-SEM and TEM

Lysine fixable biotinylated 10 kDa dextran (10 mg/ml; Thermo Fisher Scientific, Waltham, MA) was administered into the SI of ketamine/xylasine anesthetized mice. After 1 hr, mice were perfused with 4% paraformaldehyde (PFA), followed by fixation in 4% PFA overnight. Tissues were transferred to

cold PBS, embedded in 2% agarose, and sectioned using a vibratome; 100 μm thick sections were rinsed in PBS, exposed to glycine, followed by rinsing in PBS and permeabilized in 0.05% saponin. Permeabilized sections were rinsed in PBS, exposed to streptavidin-HRP (Vector Laboratories, Burlingame, CA) and exposed to Ni-DAB (Vector Laboratories, Burlingame, CA) in PBS for 10 min, followed by exposure to Ni-DAB in $H_2O_2$ for 10 min, rinsed with PBS and fixed overnight in 2% glutaraldehyde. Stained tissue specimens were embedded in resin and evaluated using a Zeiss Crossbeam 540 FIB-SEM equipped with a GEMINI II and a Capella Ga-Liquid Metal Ion Source (Ga-LMIS) column (Carl Zeiss Microscopy, Thornwood, NY). The milling process was set to 10 nm generating a dataset of ~1200 SEM images encompassing the volume of a GC. The acquired dataset was analyzed by Amira (Thermo Fisher Scientific, Waltham, MA). For the 3D visualization, organelles were manually traced and identification of the respective organelles and dextran-containing organelles was performed using Functional Ultrastructure: Atlas of Tissue Biology and Pathology (*Pavelka and Roth, 2010*). For evaluation using TEM, 50 nm sections were cut from the resin embedded tissues and sections were evaluated using a JEOL JEM-1400 120kV TEM (Jeol USA Inc, Peabody, MA). TEM sections were used to further confirm FIB-SEM findings. In the evaluated sections, 23 GCs stained positive for dextrain-containing organelles. Of these, 12 cells stained positive in the TGN, 4 cells stained positive in MVBs, 2 cells stained positive in lysosomes, and 11 cells stained positive in vesicular structures. The positive organelles were identified as structures with a clearly darker color as compared to unstained structures.

## Quantification and statistical analysis

Data is presented as mean ± standard error or the mean. Statistical analysis was performed using GraphPad Prism 7 (GraphPad Software Inc, San Diego, CA). Analyses between two groups were performed using the Student's t-test. Comparisons between three groups or more were performed using a one-way ANOVA with Dunnett's post hoc test for correction of multiple comparisons. A cut-off of $p < 0.05$ was used for statistical significance. Details regarding the statistical test used in the respective experiments are indicated in the figure legends together with populations size. All experimental groups consist of three or more technical replicates. Results of statistical tests are indicated in the figures and in the text.

## Acknowledgements

The authors gratefully acknowledge assistance in super-resolution imaging and FIB-SEM from the Washington University Center for Cellular Imaging. The authors also gratefully acknowledge the Gothenburg University Center for Cellular Imaging for assistance with confocal microscopy.

## Additional information

### Funding

| Funder | Grant reference number | Author |
|---|---|---|
| National Institute of Diabetes and Digestive and Kidney Diseases | DK097317 | Rodney D Newberry |
| National Institute of Allergy and Infectious Diseases | AI131342 | Rodney D Newberry |
| National Institute of Diabetes and Digestive and Kidney Diseases | DK109006 | Kathryn A Knoop |
| National Institute of Allergy and Infectious Diseases | AI136515 | Rodney D Newberry |
| National Institute of Allergy and Infectious Diseases | AI140755 | Rodney D Newberry |

| Funder | Grant reference number | Author |
|---|---|---|
| National Institute of Allergy and Infectious Diseases | AI112626 | Simon P Hogan |
| National Institute of Diabetes and Digestive and Kidney Diseases | DK048106 | Wayne I Lencer |
| Crohn's and Colitis Foundation | 34835 | Jenny K Gustafsson |
| Vetenskapsrådet | 2014-00366 | Jenny K Gustafsson |
| Stiftelserna Wilhelm och Martina Lundgrens | | Jenny K Gustafsson |
| Åke Wiberg Stiftelse | | Jenny K Gustafsson |
| Jeanssons Stiftelser | | Jenny K Gustafsson |

The funders had no role in study design, data collection and interpretation, or the decision to submit the work for publication.

## Author contributions

Jenny K Gustafsson, Conceptualization, Data curation, Formal analysis, Funding acquisition, Investigation, Methodology, Project administration, Resources, Supervision, Writing – original draft, Writing – review and editing; Jazmyne E Davis, Conceptualization, Data curation, Formal analysis, Funding acquisition, Investigation, Methodology, Writing – original draft, Writing – review and editing; Tracy Rappai, Data curation, Investigation, Methodology; Keely G McDonald, Data curation, Formal analysis, Methodology; Devesha H Kulkarni, Conceptualization, Data curation, Formal analysis, Funding acquisition, Methodology, Writing – original draft, Writing – review and editing; Kathryn A Knoop, Conceptualization, Data curation, Formal analysis, Funding acquisition, Writing – original draft, Writing – review and editing; Simon P Hogan, Conceptualization, Data curation, Formal analysis, Writing – original draft, Writing – review and editing; James AJ Fitzpatrick, Conceptualization, Formal analysis, Methodology, Supervision, Writing – original draft, Writing – review and editing; Wayne I Lencer, Conceptualization, Data curation, Formal analysis, Methodology, Supervision, Writing – original draft, Writing – review and editing; Rodney D Newberry, Conceptualization, Data curation, Formal analysis, Funding acquisition, Methodology, Project administration, Resources, Supervision, Writing – original draft, Writing – review and editing

## Author ORCIDs

Jenny K Gustafsson ⓘ http://orcid.org/0000-0001-7213-4065
Kathryn A Knoop ⓘ http://orcid.org/0000-0003-2007-3066
Wayne I Lencer ⓘ http://orcid.org/0000-0001-7346-2730
Rodney D Newberry ⓘ http://orcid.org/0000-0002-4152-5191

## Ethics

All animal procedures and protocols were performed in accordance with the Institutional Animal Care and Use Committee at Washington University School of Medicine (Animal Welfare Assurance number: A-3381-01) and the Swedish animal welfare legislation and approved by the Swedish Laboratory Animal Ethical Committee in Gothenburg (Ethical permit ID number: 5.8.18-11053/2019).

## Decision letter and Author response

Decision letter https://doi.org/10.7554/eLife.67292.sa1
Author response https://doi.org/10.7554/eLife.67292.sa2

---

# Additional files

## Supplementary files
• Transparent reporting form

## Data availability

All data generated and analysed for this study are included in the manuscript and source data files for figure 8.

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
