## [Editor Report]

In this study, the authors present a compelling data regarding how barrier function and immunity are coordinated within the mammalian intestinal system. This paper demonstrates that cells responsible for the secretion of the protective mucous lining of the intestine also sample substances within the intestinal lumen to present to the immune system, and how these two processes are differentially regulated.

---

## [Decision Letter]

Thank you for submitting your article "Intestinal Goblet Cells Sample and Deliver Lumenal Antigens by Regulated Endocytic Uptake and Transcytosis" for consideration by *eLife*. Your article has been reviewed by 3 peer reviewers, one of whom is a member of our Board of Reviewing Editors, and the evaluation has been overseen by a Reviewing Editor and Vivek Malhotra as the Senior Editor. The reviewers have opted to remain anonymous.

Essential revisions:

1) Additional immunolabeling data (mACh subtypes, different endocytic markers)

2) Enhanced presentation of the FIB SEM data

3) Images of the intestinal epithelium following drug treatment

*Reviewer #1:*

The authors examined how two processes (antigen sampling of the intestinal environment and mucus secretion to protect the intestine) are regulated in goblet cells of the small intestine and colon. They find differential regulation of these two processes and that they are not functionally linked. This work involves pharmacological investigation of signaling pathways and calcium regulation, demonstrating distinct responses. The understanding of how these two processes are regulated provides a more detailed understanding of how barrier maintenance and antigen delivery to the immune system are coordinated within the digestive system.

1. I would recommend reorganizing the panels in Figure 5 to pair the results of each cell type for GAP formation vs. mucus secretion so that they are more easily compared. For example, panel A would be SI villus secretion and panel B would be SI villus GAP formation.

2. Are there any data indicating whether there are differences in the composition of the center vs. lateral regions the apical membrane of goblet cells to provide insight into how these two processes are spatially segregated?

*Reviewer #2:*

The authors investigate the formation and cell biology of goblet cell associated antigen passages (GAPs). Using FIB SEM and a fluid phase tracer they provide a 3D model of the structure of a goblet cell including identification of key parts of the trafficking pathway. By TEM they also provide evidence of vesicle uptake by underlying cells. Using pharmacological inhibitors, they provide the first evidence of differential machinery for independent regulation of GAP formation separate from the exocytic machinery necessary for mucin granule exocytosis. They conclude that GAP formation is a transcytosis process based on endocytosis at the edge of the apical side of the goblet cell (in an EEA1 independent manner) followed by trafficking throughout the cytoplasm before exocytosis at the basal side.

This is a convincing study providing good evidence of related but separately regulated endocytic and exocytic processes to mediate important physiologically and pathologically relevant functions.

Strengths

The major strength of the manuscript is the in vivo imaging aspects in mice. These provide analysis of the phenomena in situ, adjacent to the appropriate associated gut cells, enterocytes etc. It also allows analysis of the phenomena in discrete sections of the intestinal tract. The quantification of phenomena is robust, careful and thorough and appropriately statistically powered.

The study provides good evidence for differences in regulation of GAP formation vs mucin granule exocytosis both in terms of their activation by muscarinic receptors and the downstream signalling from extracellular and intracellular calcium stores. This supports the conclusion that both of these phenomena although related can be differentially controlled. The rationale for the necessity for separate pathways existing is well thought out and logical.

The FIB SEM electron microscopy analysis with labelled streptavidin provides a clear and striking delineation of the trafficking route. It convincingly highlights all the structures the authors noted and supports the assertion that a transcytosis route provides a means to safely sample fluid phase material from the gut lumen.

Weaknesses

The experiments rely largely on pharmacological interventions to monitor the formation of GAPs and to pick apart the differential regulation of GAP formation vs mucin secretion. These agents will impinge on many intracellular pathways occurring in the cell making effects on GAP formation and mucin secretion potentially both direct and indirect.

Images of the effect of some of the drugs and chemicals on goblet cells as they undergo GAP formation and mucin release are likely to be informative as to the direct or indirect nature of the exposure. If an arrest of trafficked dextran is noted at certain points on the surface of the goblet cell, or at discrete points within the cell following dynamin, actin or microtubule inhibition this would indicate more specifically the point at which these pieces of molecular machinery are required. Although exposure of the inner surface of the gut to EGTA is likely to interfere with traffic of Calcium from an extracellular source it is also likely to disrupt intracellular junctions which may have gross and potentially indirect effects.

The article is very well written, clear and the subject matter is important.

It would be helpful to include more images to support the quantification of the affects you note in Figure 1D. Such images might shed light on the specific role of the inhibited pathways, if for example dextran labelled pools accumulate near the surface, Golgi etc. It will also provide a better idea as to how the drug treatments are more broadly affecting the cells in question.

The volume electron microscopy images shown in figure 2 are striking. A movie might help give an idea as to how well the segmentation of the em images fits with the rendered images shown, it might also provide better context to the arrangement of the structures in the goblet cell. Are labelled vesicles present towards the edge of the cell? Are there examples of vesicles elsewhere perhaps in amongst the mucin granules? Also, is it possible to catch endocytic structures forming at the apical edge of the cell as posited.

Additional markers of the endocytic pathway associated with GAP formation would also be beneficial. The authors convincingly showed EEA1 is not present, it would be helpful to compare localisation of late endosomes, lysosomes, recycling endosomes, the ER and the TGN. Especially as some of these structures are thought to provide a reservoir for calcium.

Associated images of the cells exposed to the calcium related drugs would also be helpful, again to get an idea as to if there are any intermediates of GAPs and to get an idea as to the health of the tissue. The BAPTA-AM appears to actively reduce the numbers of GAPs… Are the effects over the vehicle significant and does this provide further information about their stability and formation?

There are typographical errors on line 694 and 760.

Generally, this is a well-conceived and interesting study providing good evidence of the cell biology of GAP formation and mucin granule secretion in the goblet cells of mice.

*Reviewer #3:*

Combined two-photon in vivo imaging, super-resolution imaging, and FIB-SEM, these authors tried to follow the fate of the fluorescent tracer injected to the gut lumen. They suggested the identity of GAP as results of ACh stimulated endocytic events that are retrograded cross the Golgi network for transcytosis on the other side of the goblet cells. Besides, ACh induced GAP formation and mucin secretion via distinct receptors and signallng pathways.

Major strengths: Detailed phamacological experiments clearly established two distinct pathways of ACh induced GAP formation and mucin secretion, which could lead to better understanding of the connection between goblet physiology and intestinal immunity.

Weaknesses:

1. It is unclear why ACh induces both GAP formation and mucin secretion in the same goblet cell. What would be the benefit of such arrangements?

2. The current data are based mostly on phamacological perturbation. I wonder whether the authors could at least show some non-overlapping distributions of mAChR1, mAChR2, mAChR3, and mAChR4.

3. Based on imaging data, the authors concluded the transcytosis fate of GAP vesicles. However, partial colocalization of dextran with the trans-Golgi marker and its fuzzy co-localization with Rab3D conferred the evidence, which is weak. We may need more direct evidence to better support this assertion.

4. Indeed, the authors have used sophisticated in vivo imaging techniques. However, they did not show the processes of dextran internalization in vivo or any time-dependent changes, which did not seem to fully use the technology's potential. In addition, some images were fuzzy, and no structures can be identified.

5. No mention of the blue labels in Figure 1, which I presumably to believe to be nulceus. If that is true, I wonder why red OVA puncta were in the middle of blue nucleus (Figure 1C).

6. In addition, in Figure 1H, the authors claimed little overlapping between dextran and EEA1. However, structures highlighted by the arrows seem to be dual labeled with both markers.

7. The authors have stated that they have obtained four cells underwent FIB-SEM imaging and used one as the example and the left as the replicates to confirm. I would suggest that they shall have some critera/quantifications that are indeed confirmed by the data not shown.

---

## [Author Response]

Essential revisions:1) Additional immunolabeling data (mACh subtypes, different endocytic markers)

Labeling of mAChR subtypes: Figure 5—figure supplement 1 and 2. Labeling of endocytic markers: Figure 2—figure supplement 1, and Figure 6—figure supplement 1.

2) Enhanced presentation of the FIB SEM data

Enhanced presentation of the FIB-SEM data: Video 1 and 2.

3) Images of the intestinal epithelium following drug treatment

Images of the epithelium following drug treatment: Figure 1—figure supplement 1, Figure 6E-K, and Figure 7C-I.

Reviewer #1:The authors examined how two processes (antigen sampling of the intestinal environment and mucus secretion to protect the intestine) are regulated in goblet cells of the small intestine and colon. They find differential regulation of these two processes and that they are not functionally linked. This work involves pharmacological investigation of signaling pathways and calcium regulation, demonstrating distinct responses. The understanding of how these two processes are regulated provides a more detailed understanding of how barrier maintenance and antigen delivery to the immune system are coordinated within the digestive system.1. I would recommend reorganizing the panels in Figure 5 to pair the results of each cell type for GAP formation vs. mucus secretion so that they are more easily compared. For example, panel A would be SI villus secretion and panel B would be SI villus GAP formation.

We agree with the reviewer and have reorganized Figure 5 according to the suggestions and updated the text to fit the new figure layout. Page 17.

2. Are there any data indicating whether there are differences in the composition of the center vs. lateral regions the apical membrane of goblet cells to provide insight into how these two processes are spatially segregated?

To our knowledge there are no studies demonstrating differences in the organization of the apical membrane. In goblet cells, the ER is surrounding the theca and one explanation to the spatial segregation of these processes is that they depend on, and access, different pools of calcium.

Reviewer #2:[…]The article is very well written, clear and the subject matter is important.It would be helpful to include more images to support the quantification of the affects you note in Figure 1D. Such images might shed light on the specific role of the inhibited pathways, if for example dextran labelled pools accumulate near the surface, Golgi etc. It will also provide a better idea as to how the drug treatments are more broadly affecting the cells in question.

We agree with the reviewers and have added images to support the quantification. The added images are found in Figure 1—figure supplement 1, indicated on page 5 lines 115, 118, 120 and 121. We have not found pools of dextran accumulating near the golgi, thus the inhibition appears to be occurring at the initial uptake event. In rare instances we have observed dextran accumulating at the surface of the cell but the events are not common enough for us to quantify.

The volume electron microscopy images shown in figure 2 are striking. A movie might help give an idea as to how well the segmentation of the em images fits with the rendered images shown, it might also provide better context to the arrangement of the structures in the goblet cell.

We agree with the reviewer and have added 2 Videos to better visualize the 3D mode and the segmentation. Video 1 shows the rendered 3D model, and Video 2 shows the segmentation that the model is based on the EM data.

Are labelled vesicles present towards the edge of the cell? Are there examples of vesicles elsewhere perhaps in amongst the mucin granules? Also, is it possible to catch endocytic structures forming at the apical edge of the cell as posited.

The labelled vesicles are mainly found at the edge of the cell and a few vesicles are found among the mucin granules. The distribution of vesicles within the cells is now visualized in a clearer way in Video 2, added to the revised manuscript, which demonstrates a Z stack of the FIB-SEM images and the corresponding Z stack of the segmentation that the model is based on. We have observed vesicles containing lumenal tracer at the apical edge of the GAP as shown in Figure 1C, and in the ultrastructural model. We have not captured the initial event of GAP formation in these studies, in part due to the experimental setup evaluating tissue sections one hour after administration of lumenal tracers, and in part because our studies focused on goblet cells definitively forming GAPs, identified by complete filling with lumenal tracer, as opposed to goblet cells potentially endocytosing lumenal substances via other pathways or outcomes.

Additional markers of the endocytic pathway associated with GAP formation would also be beneficial. The authors convincingly showed EEA1 is not present, it would be helpful to compare localisation of late endosomes, lysosomes, recycling endosomes, the ER and the TGN. Especially as some of these structures are thought to provide a reservoir for calcium.

We agree with the reviewer and have added staining of the late endosome marker Rab7, the lysosome marker LAMP1, the TGN marker TGN46, and the ER marker calnexin to the revised manuscript. These findings are shown in Figure 2—figure supplement 1 on page 32, and Figure 6—figure supplement 1 on page 35 and referenced in the text on page 9 lines 197 -199, 205-209 and on page 20 line 446-451.

Associated images of the cells exposed to the calcium related drugs would also be helpful, again to get an idea as to if there are any intermediates of GAPs and to get an idea as to the health of the tissue. The BAPTA-AM appears to actively reduce the numbers of GAPs… Are the effects over the vehicle significant and does this provide further information about their stability and formation?

We agree with the reviewer and have split Figure 6 into Figure 6 on page 19 and Figure 7 on page 21 to include visualization of the effect that the respective inhibitors have on the tissue. The BAPTA-AM treatment does indeed reduce GAP numbers below baseline, demonstrating that both spontaneous and induced GAP formation is dependent on intracellular ca^2+^. We have added information regarding this in the revised manuscript, page 18, lines 4010 -413. We have not observed intermediates of GAP formation following treatment with the inhibitors. We interpret this to indicate that the inhibitors act on early events in GAP formation, or these events are below the limit of detection of our assays.

There are typographical errors on line 694 and 760.

We apologize and have corrected the typographical error in the revised manuscript.

Reviewer #3:Combined two-photon in vivo imaging, super-resolution imaging, and FIB-SEM, these authors tried to follow the fate of the fluorescent tracer injected to the gut lumen. They suggested the identity of GAP as results of ACh stimulated endocytic events that are retrograded cross the Golgi network for transcytosis on the other side of the goblet cells. Besides, ACh induced GAP formation and mucin secretion via distinct receptors and signallng pathways.Major strengths: Detailed phamacological experiments clearly established two distinct pathways of ACh induced GAP formation and mucin secretion, which could lead to better understanding of the connection between goblet physiology and intestinal immunity.Weaknesses:1. It is unclear why ACh induces both GAP formation and mucin secretion in the same goblet cell. What would be the benefit of such arrangements?

We can only speculate on the benefit or purpose of such an arrangement. Since the inception of this project four years ago, our thinking on the cellular basis of GAP formation has evolved significantly, as reflected by this work and paralleled by the reviewer’s comment. We too were surprised that GAP formation and secretion were induced by the same stimulus, ACh, and while the processes of secretion and membrane retrieval are inherently linked, GAP formation and ACh induced mucus secretion are not functionally linked or dependent upon each other, and yet can occur in the same cell. We interpret our observations to indicate that GAP formation evolved as an additional, and redundant, pathway of membrane recovery that is specialized to deliver luminal substances to the transcytotic pathway. In other work we have observed that while exogenous ACh (analogues) can augment GAP formation, ACh is generally not limiting for GAP formation and GAP formation is largely regulated by inhibiting AChR responsiveness in goblet cells. Incorporating these observations would suggest that goblet cells evolved in an efficient manner to use an available stimulus and the machinery of a highly secretory cell to perform two related but independent tasks allowing them to independently maintain the barrier and/or perform surveillance of the luminal contents. However, we would note that goblet cells may be heterogeneous in their functions, which will require additional studies to dissect.

2. The current data are based mostly on phamacological perturbation. I wonder whether the authors could at least show some non-overlapping distributions of mAChR1, mAChR2, mAChR3, and mAChR4.

We agree with the reviewers. Immunostaining for the respective receptors using currently available antibodies was unsuccessful. However, we were able to identify mAChR expression using RNAscope for mAChR1, 3 and 4. We did not stain for mAChR2 since this receptor subtype is not discussed in this manuscript. This data is now included in Figure 5—figure supplement 1 on page 33 and Figure 5—figure supplement 2 on page 34 and discussed in the revised manuscript on page 16, lines 372-384.

3. Based on imaging data, the authors concluded the transcytosis fate of GAP vesicles. However, partial colocalization of dextran with the trans-Golgi marker and its fuzzy co-localization with Rab3D conferred the evidence, which is weak. We may need more direct evidence to better support this assertion.

We agree. Rab3D (Figure 2C, D) as well as VAMP8 (Figure 2—figure supplement 1D) staining that has been added to the revised manuscript. These markers co-localized with dextran in the lateral and supranuclear portions of goblet cells forming GAPs, but do not extend to vesicles at the basolateral region, suggesting that Rab3D and VAMP8 marks fates of vesicles containing luminal substances proximal to the transcytosis event. The transcytosis fate of the vesicles occurring when GAPs form is supported by the FIB-SEM studies in Figure 2 on page 10, including the TEM images in Figure 2F and G, and the super resolution microscopy in Figure 1C on page 7, which are as granular as these approaches allow. These studies along with prior work demonstrating that LP-APCs acquire both luminal substances and goblet cell proteins from GAPs (McDole et al., *Nature*, 2012) have led us to conclude that the GAP vesicles are transcytosed. However, we acknowledge this concern and are cautious with our interpretation of events at the basolateral surface of GAPs resulting in the transfer of lumenal substances, which will require further studies to fully comprehend.

4. Indeed, the authors have used sophisticated in vivo imaging techniques. However, they did not show the processes of dextran internalization in vivo or any time-dependent changes, which did not seem to fully use the technology's potential. In addition, some images were fuzzy, and no structures can be identified.

In prior work we have demonstrated GAP formation using in vivo two photon imaging of live anesthetized mice. These time lapse images demonstrate that GAPs form from the apical to basolateral surface through the acquisition of luminal substances (McDole et al. *Nature*, 2012). However, these studies suffer from the relative low resolution of two photon microscopy that led to the initial incorrect assumptions regarding GAP formation. All studies in this manuscript were performed in vivo and analyzed on fixed tissue sections, with the exception of Figure 1A which is an image from in vivo two photon imaging. As noted previously, the inhibitors used in this study act on GAP formation at early stages, and therefore did not allow us to evaluate intermediate events. In addition, the focus of these studies was on the process of GAP formation, as opposed to events at the apical or basolateral surfaces, so we chose to evaluate tissues after one hour of incubation with luminal dextran, a time point which we have previously observed that most GAPs are fully formed. Further, we acknowledge that the definition of a GAP is delivery of luminal substances to the basolateral surface, and as such delineation of early GAP events vs. other endocytic events recovering membranes, which the literature indicates exists to maintain the secretory potential of highly secretory cells, would be very difficult. In summation, we agree with the reviewer’s comments and acknowledge this limitation, however we believe this work is a solid step forward to delineating GAP formation from other processes in goblet cells.

5. No mention of the blue labels in Figure 1, which I presumably to believe to be nulceus. If that is true, I wonder why red OVA puncta were in the middle of blue nucleus (Figure 1C).

We apologize and have corrected this in the revised version of the manuscript. Figure 1C represents a 3D projection of a z-stack such that the red puncta may be overlaid upon the nucleus. Further it is worth noting that nuclei are not smooth, as can be seen in the FIB-SEM reconstructions and the red puncta may be interspersed between folds of the nucleus. Thus, the red puncta are located near but not in the nucleus. We have added information regarding this in the revised manuscript page 8 line 170.

6. In addition, in Figure 1H, the authors claimed little overlapping between dextran and EEA1. However, structures highlighted by the arrows seem to be dual labeled with both markers.

We apologize for the lack of clarity. The dual labeling is observed in the enterocytes adjacent to the goblet cells but not in the goblet cells. The text page 6, lines 145-146 have been revised for clarity.

7. The authors have stated that they have obtained four cells underwent FIB-SEM imaging and used one as the example and the left as the replicates to confirm. I would suggest that they shall have some critera/quantifications that are indeed confirmed by the data not shown.

We agree. We obtained four complete renderings of goblet cells undergoing GAP formation by FIB-SEM. The process of FIB-SEM of a goblet cell requires ~72 hours of continuous imaging (~1,200 SEM images) and some attempts to obtain an entire goblet cell were not successful, thus this information does not represent the entirety of the experiments performed. The process of constructing a 3D model of a GAP from a single FIB-SEM experiment required months of computer processing. Therefore, we chose one goblet cell undergoing GAP formation to generate our 3D model, and multiple FIB-SEM and TEM images to confirm this model. We have added a video (Video 2) demonstrating the Z stack of the FIB-SEM images with the corresponding Z stack of the segmented data to better convey the model. Additional information regarding the confirmation of the FIB-SEM data has been added to the Results section, page 8 line 186-187, and the methods section, page 43 lines 914-916 and page 44 lines 917-919.